# Structural basis for mTORC1 activation on the lysosomal membrane

Zhicheng Cui[1,2], Alessandra Esposito[3,4], Gennaro Napolitano[3,4,5], Andrea Ballabio[3,4,5,6,7] & James H. Hurley[1,2,8]✉

The mechanistic target of rapamycin complex 1 (mTORC1) integrates growth factor (GF) and nutrient signals to stimulate anabolic processes connected to cell growth and inhibit catabolic processes such as autophagy[1,2]. GF signalling through the tuberous sclerosis complex regulates the lysosomally localized small GTPase RAS homologue enriched in brain (RHEB)[3]. Direct binding of RHEB–GTP to the mTOR kinase subunit of mTORC1 allosterically activates the kinase by inducing a large-scale conformational change[4]. Here we reconstituted mTORC1 activation on membranes by RHEB, RAGs and Ragulator. Cryo-electron microscopy showed that RAPTOR and mTOR interact directly with the membrane. Full engagement of the membrane anchors is required for optimal alignment of the catalytic residues in the mTOR kinase active site. Converging signals from GFs and nutrients drive mTORC1 recruitment to and activation on lysosomal membrane in a four-step process, consisting of (1) RAG–Ragulator-driven recruitment to within ~100 Å of the lysosomal membrane; (2) RHEB-driven recruitment to within ~40 Å; (3) RAPTOR–membrane engagement and intermediate enzyme activation; and (4) mTOR–membrane engagement and full enzyme activation. RHEB and membrane engagement combined leads to full catalytic activation and structurally explains GF and nutrient signal integration at the lysosome.

RHEB is required for phosphorylation of the canonical substrates of mTORC1 (ref. 5), including the ribosomal protein S6 kinase (S6K) and eukaryotic translation initiation factor 4E-binding protein 1 (4E-BP1) that mediate mTORC1 stimulation of protein synthesis[1], whereas it is dispensable for the phosphorylation of noncanonical substrates, such as transcription factor EB (TFEB)[6]. RHEB is present in cells at an estimated concentration[7] of 650 nM, yet >100 μM soluble RHEB–guanosine triphosphate (GTP) is needed for half-maximal mTORC1 activation in solution[4]. Thus, the discrepancy between physiological concentrations and the in vitro biochemistry of RHEB, the most fundamental activator of mTORC1, is greater than two orders of magnitude. RHEB is farnesylated[8–10], which is essential for mTORC1 activation and is responsible for targeting RHEB to lysosomes. Despite the fact that the receptor–PI3K–AKT signalling pathway originates at the plasma membrane, RHEB signalling to mTORC1 occurs exclusively on the cytosolic face of lysosomes[3]. mTORC1 is recruited to lysosomes by the RAS-related GTP-binding (RAG) GTPases RAGA–RAGD under amino acid-replete conditions[11–13]. The RAGs function as heterodimers, with RAGA or RAGB paired with RAGC or RAGD. Active RAG dimers consisting of RAGA–GTP or RAGB–GTP and RAGC–guanosine diphosphate (GDP) or RAGD–GDP recruit mTORC1 to the lysosome by binding to its RAPTOR subunit[14,15]. The RAGs, in turn, are recruited to the lysosomal membrane by the pentameric Ragulator complex, specifically, by its myristoylated and palmitoylated LAMTOR1 subunit[16]. Amino acid-dependent mTORC1 recruitment to lysosomes by the RAGs and Ragulator brings mTORC1 in proximity to the lysosome-bound pool of RHEB–GTP. The dual dependence on lysosomal localization and RHEB engagement serves as a logical 'AND' gate for a GF signal and an ample pool of biosynthetic precursors before protein synthesis and cell growth. How this physiologically critical AND gate might be organized and implemented at the structural level is unknown.

We proposed that the lysosomal membrane itself might be the missing link that orchestrates the AND gate and compensates for the more-than-two orders of magnitude in concentration needed for RHEB–GTP activation of mTORC1. The feasibility of atomistic single-particle cryo-electron microscopy (cryo-EM) reconstructions of liposome-bound peripheral protein assemblies has recently been demonstrated[17,18]. This prompted us to reconstitute the concerted activation of mTORC1 on liposomes by RHEB and RAGA–RAGC–Ragulator and elucidate its structural basis.

## mTORC1 activation on membranes

We used large unilamellar vesicles (LUVs), with a lipid composition of 72.8% DOPC, 7% POPS, 10% cholesterol, 5% DGS-nitrilotriacetic acid (NTA), 5% PE MCC and 0.2% DiR, as the membrane platform to investigate the role of RAG–Ragulator and RHEB in mTORC1 activation. The active RAGA$^{GTP}$–RAGC$^{GDP}$–Ragulator complex was recruited to LUVs

[1]Department of Molecular and Cell Biology, University of California, Berkeley, Berkeley, CA, USA. [2]California Institute for Quantitative Biosciences, University of California, Berkeley, Berkeley, CA, USA. [3]Telethon Institute of Genetics and Medicine, Naples, Italy. [4]Genomics and Experimental Medicine program, Scuola Superiore Meridionale, School for Advanced Studies, Naples, Italy. [5]Medical Genetics Unit, Department of Medical and Translational Science, Federico II University, Naples, Italy. [6]Department of Molecular and Human Genetics, Baylor College of Medicine, Houston, TX, USA. [7]Jan and Dan Duncan Neurological Research Institute, Texas Children's Hospital, Houston, TX, USA. [8]Helen Wills Neuroscience Institute, University of California, Berkeley, Berkeley, CA, USA. ✉e-mail: jimhurley@berkeley.edu

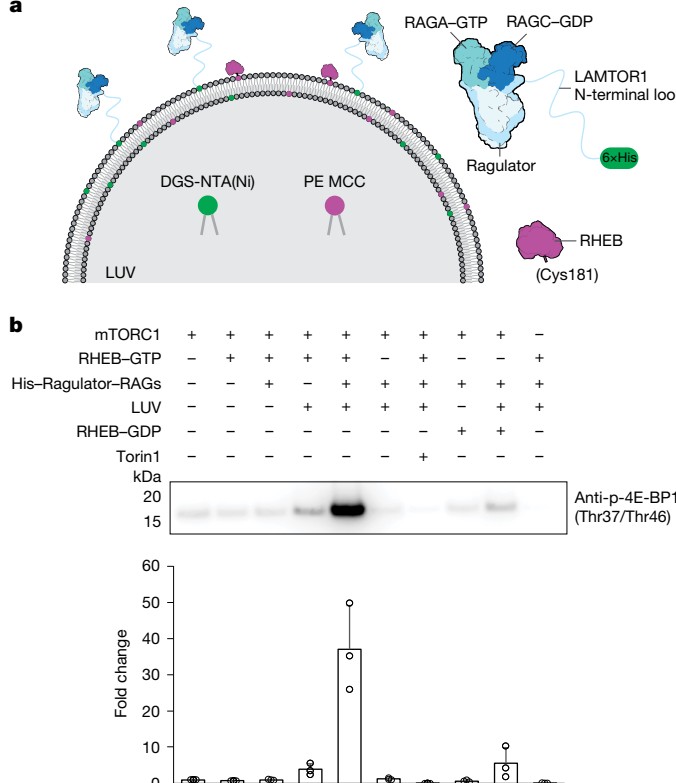

**Fig. 1 | In vitro reconstitution and mTORC1 kinase activity with liposomes.**
**a**, Cartoon showing the in vitro membrane-containing reconstitution of mTORC1 kinase activity. N-terminal 6×His LAMTOR1 is tethered with DGS-NTA(Ni), and Cys181 of RHEB is lipidated by reaction with PE MCC. **b**, Determinants of mTORC1 activation. Western blot is shown for anti-phospho (p)-4E-BP1 (Thr37/Thr46). Activation fold change is computed relative to the activity of mTORC1 alone. Quantifications are indicated with bars from three experiments and are mean ± s.d. All samples were derived from the same experiment, and gels and blots were processed in parallel.

containing the lipid DGS-NTA(Ni) by means of a 6×His tag fused to the N terminus of the LAMTOR1 subunit of Ragulator in place of the physiological myristoyl and palmitoyl modifications (Fig. 1a). RHEB was tethered to the LUVs by means of a thiol–maleimide reaction between the functionalized lipid PE MCC and its only cysteine residue, C181, which is the farnesylation site of endogenous RHEB (Fig. 1a). RHEB and RAGA–RAGC–Ragulator were used at essentially physiological concentrations of 250 and 300 nM, respectively. We monitored mTORC1 kinase activity as a function of LUVs, RHEB and RAG–Ragulator by detecting Thr37 and Thr46 phosphorylation of full-length 4E-BP1. mTORC1 kinase activity increased more than 35-fold in the presence of LUVs, RAG–Ragulator and RHEB–GTP (Fig. 1b) but not with RHEB–GDP. As expected, no activity was observed in the presence of the mTOR inhibitor Torin1 or in the absence of mTORC1. LUVs and RHEB–GTP increased mTORC1 activity by about threefold, whereas other combinations did not affect its activity. No increase in mTORC1 activity was observed when 250 nM RHEB–GTP was present but LUVs were absent, consistent with a previous report that >100 μM soluble RHEB–GTP is required for activation[4]. Therefore, the combination of liposomes and liposome-tethered RHEB and RAGA–RAGC–Ragulator synergistically and potently activates mTORC1 phosphorylation of 4E-BP1.

## mTORC1 structure on a membrane

We reconstituted mTORC1–RHEB–RAG–Ragulator with the full-length substrate 4E-BP1 complex on liposomes and acquired cryo-EM images

(Fig. 2a). Two-dimensional (2D) class averages suggested that the complex was rigidly oriented with respect to the phospholipid bilayer (Fig. 2b). We determined the cryo-EM structure of the entire assembly to an overall resolution of 3.23 Å (Extended Data Fig. 1 and Table 1). The overall structure of mTORC1 and its interactions with RHEB and RAG–Ragulator are consistent with previous structures determined in the absence of membranes[4,14,15,19,20], including the large conformational change induced by RHEB binding[4]. However, when superimposing the membrane-bound mTORC1–RHEB structure with the soluble mTORC1–RHEB structure on the basis of the mTOR subunit, we observed a conformational change in the RAPTOR subunit, with a rotation of about 7° towards the HEAT domain of mTOR (Extended Data Fig. 2a). This rotational movement of RAPTOR in membrane-bound mTORC1–RHEB aligns with the transition from apo-mTORC1 to soluble mTORC1–RHEB, suggesting that the membrane plays a part in further constricting mTORC1 when bound to RHEB. Local refinement of the mTOR–RHEB–MLST8 and RAPTOR–RAG–Ragulator subcomplexes yielded cryo-EM density maps at resolutions of 3.12 Å and 2.98 Å, respectively, allowing us to build atomically detailed models of the entire assembly (Fig. 2c,d). The high-resolution cryo-EM density of nucleotides confirmed the active states of RAG GTPases (Extended Data Fig. 2b,c). The density of inositol hexakisphosphate was observed, surrounded by the lysine–arginine cluster in the FAT domain of mTOR as previously reported in mTORC2 structures[21] and the TFEB-containing megacomplex of mTORC1 (ref. 20) (Fig. 2e). Despite the inclusion of full-length 4E-BP1, only the TOR signalling motif was visualized (Fig. 2f), bound to the same location on the RAPTOR subunit as previously identified[4,22].

We noticed two three-dimensional (3D) classes with extra RAG–Ragulator densities bound to the MLST8 subunit of mTORC1. These were refined to overall resolutions of 3.47 Å and 3.81 Å for classes with the one and two extra copies of RAG–Ragulator, respectively (Extended Data Fig. 3). We pooled these classes together and carried out a focused refinement of the MLST8–RAG–Ragulator subcomplex, yielding a cryo-EM map with a resolution of 4 Å. The structure showed direct interactions between MLST8 and RAGA–GTP. Residues His49 and Arg51 of the interswitch region of RAGA maintained electrostatic interactions with an acidic patch on MLST8, involving residues Asp181, Asp213 and Glu300 (Extended Data Fig. 4a,b). The ordered interswitch region of RAGA in the GTP-bound state ensures that only RAGA–GTP can interact with MLST8 (Extended Data Fig. 4c). This previously unknown RAG–Ragulator site partially overlaps the mSIN subunit of mTORC2, consistent with the absence of interaction between RAG–Ragulator and mTORC2 (Extended Data Fig. 5a). This site is near the binding site of the mTORC1 inhibitory protein PRAS40 (ref. 4) (Extended Data Fig. 5b).

## mTOR–membrane and RAPTOR–membrane interactions

The cryo-EM reconstruction showed that mTOR and RAPTOR subunits directly interact with the membrane, in addition to the expected membrane attachment by lipidated LAMTOR1 and RHEB (Fig. 3a). In line with previous publications[14,15,23–25], the N-terminal membrane-anchoring segment of LAMTOR1 was not visualized. The last ordered residue, Ala173 of RHEB, is in the middle of the hypervariable domain, indicating a partially folded hypervariable domain and a flexible eight-residue loop attached to the membrane through Cys181, which places RHEB about 13 Å above the membrane (Fig. 3b). The hydrophobic side chains Phe1296 and Met1297 in the WD40 domain of RAPTOR, which we denote as the FM finger, manifested membrane interactions in the cryo-EM density map (Fig. 3c). Specific residues in the N-HEAT domain of mTOR, namely Lys471, Arg472 and Lys474, form a basic loop that is also in close contact with the membrane (Fig. 3d). The entire active mTORC1 assembly showed a large membrane-covering footprint (Fig. 3e), owing to the membrane contact sites of the mTOR and RAPTOR subunits and the small membrane–RHEB gap (Fig. 3b).

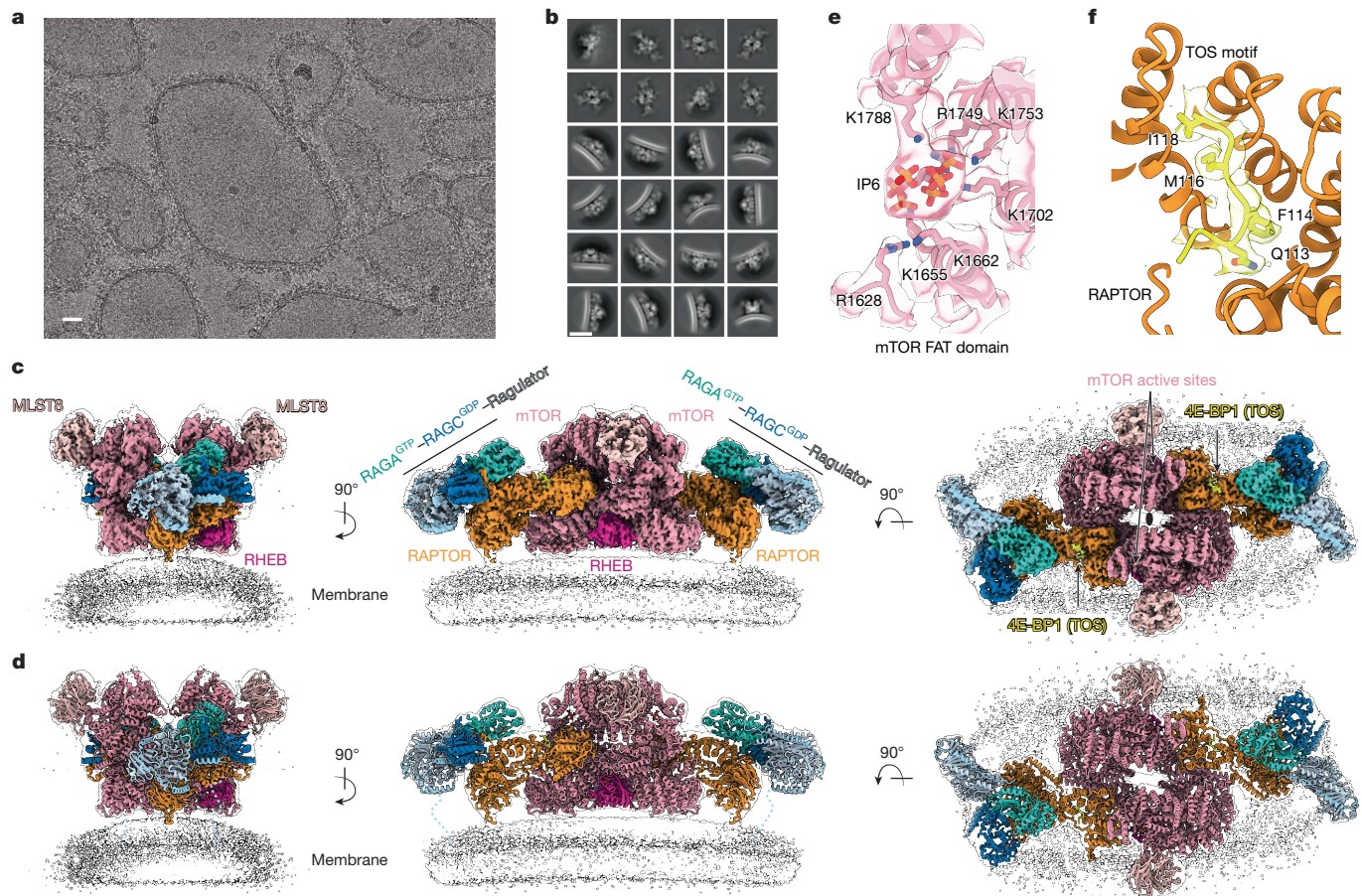

**Fig. 2 | Cryo-EM structure of mTORC1–RHEB–RAG–Ragulator–4E-BP1 on the membrane. a**, Representative example chosen from 29,301 cryo-EM images showing protein-decorated liposomes. **b**, Representative 2D averages showing side and top views of the protein–membrane complex. **c**, A composite cryo-EM density map of mTORC1–RHEB–RAG–Ragulator–4E-BP1 on a membrane, assembled from two focused refinement maps (mTOR–RHEB–MLST8 and RAPTOR–RAG–Ragulator), overlaid with the unsharpened cryo-EM map (contour level, 0.033) from the overall refinement with *C*2 symmetry.

The active sites of mTOR are labelled with arrows. The twofold axis is labelled as a black oval symbol in the top view. Different contour levels were used for optimal visualization using UCSF ChimeraX. **d**, Atomic model of mTORC1–RHEB–RAG–Ragulator–4E-BP1 is overlaid with the unsharpened cryo-EM map from the overall refinement with *C*2 symmetry. **e**, Close-up view of the density for inositol hexakisphosphate (IP6) (contour level, 0.156) and the surrounding lysine–arginine cluster. **f**, Close-up view of the density for the 4E-BP1 TOR signalling (TOS) motif (contour level, 0.097). Scale bars (**a**,**b**), 20 nm.

The two RAPTOR–membrane contact sites in the dimeric structure are separated by 230 Å (Fig. 3e). To accommodate both contact sites simultaneously, the liposome diameter must be large enough to provide a relatively flat platform within the potential flexibility range of the complex. Accordingly, we reasoned that liposomes with smaller diameters would result in reduced mTORC1 activation. Liposomes with mean diameters of 67–81 nm showed an approximate 65% reduction in mTORC1 kinase activity compared with those with a mean diameter of 355 nm (Fig. 3f), bearing out the structural prediction. After reading a report that phosphatidylinositol 3-phosphate and phosphatidylinositol 4-phosphate have opposing effects on mTORC1 activity in cells[26], we tested these lipids in the in vitro kinase assay (Extended Data Fig. 6a) and found no difference in activity between these two phosphoinositides and another anionic lipid, phosphatidylserine. These data show that membrane curvature, but not phosphoinositide identity, directly modulates mTORC1 activity on the membrane surface.

To validate the biochemical role of the membrane-interacting residues of RAPTOR and mTOR, we mutated RAPTOR residues Phe1296 and Met1297 and mTOR residues Lys471, Arg472 and Lys474 to acidic residues to maximally disrupt membrane anchoring. We then purified mTORC1 with either the RAPTOR mutations, the mTOR mutations or a combination of both and tested them using an in vitro kinase assay. Mutations in mTORC1 containing RAPTOR[F1296E/M1297E] alone, mTOR[K471D/R472D/K474D]

alone or the combination of both reduced mTORC1 kinase activity (Fig. 3g), consistent with the membrane-docked structure. To validate the functional role of the RAPTOR–membrane interaction, we tested the RAPTOR[F1296E] and RAPTOR[F1296E/M1297E] constructs in inducible RAPTOR knockout mouse embryonic fibroblasts (MEFs) (Fig. 3h). After treatment with 4-hydroxytamoxifen, we confirmed RAPTOR knockout. We observed a significant reduction in 4E-BP1 and S6K phosphorylation in both RAPTOR mutants compared with the wild type upon amino acid replenishment. As expected, given that these RAPTOR mutants are competent to bind RAGA, lysosomal mTOR localization was not affected by the membrane-anchoring mutations (Extended Data Fig. 6b). These results collectively show that the full activation, but not lysosomal localization, of mTORC1 requires interactions between the membrane and the mTOR and RAPTOR subunits.

## RHEB activation of mTORC1 on the membrane

Using symmetry expansion and particle subtraction to remove RAPTOR–RAG–Ragulator, we performed 3D classification on the subpopulation that does not contain extra RAG–Ragulator copies and identified two distinct conformations of the mTOR–RHEB–MLST8 subcomplex. We designate these states as intermediate and fully active for reasons described below. The final resolutions of the

**Table 1 | Cryo-EM data collection, refinement and validation statistics**

| | mTORC1–RAG–Ragulator–4E-BP1 on membrane (EMD-47932) (PDB 9ED4) | mTOR intermediate state on membrane (EMD-47940) (PDB 9ED8) | mTOR active state on membrane (EMD-47939) (PDB 9ED7) | RAG–Ragulator–MLST8 (EMD-47933) (PDB 9ED6) |
|---|---|---|---|---|
| **Data collection and processing** | | | | |
| Magnification | 81,000 | 81,000 | 81,000 | 81,000 |
| Voltage (kV) | 300 | 300 | 300 | 300 |
| Electron exposure (e⁻/Å²) | ~30 | ~30 | ~30 | ~30 |
| Defocus range (μm) | −0.9 to −2.0 | −0.9 to −2.0 | −0.9 to −2.0 | −0.9 to −2.0 |
| Pixel size (Å) | 1.05 | 1.05 | 1.05 | 1.05 |
| Symmetry imposed | C2 | C2 symmetry expanded, C1 | C2 symmetry expanded, C1 | C2 symmetry expanded, C1 |
| Initial particle images (no.) | 54,935,004 (blob picking) | 54,935,004 (blob picking) | 54,935,004 (blob picking) | 54,935,004 (blob picking) |
| Final particle images (no.) | 359,012 | 109,105 | 133,193 | 190,751 |
| Map resolution (Å) | 3.23 overall (3.12/2.98 local refine) | 3.16 | 3.61 | 3.98 |
| FSC threshold | 0.143 | 0.143 | 0.143 | 0.143 |
| Map resolution range (Å) | ~2.3 to ~38 ~2.3 to ~40.2 | ~2.3 to ~41.9 | ~2.2 to ~41.2 | ~3.37 to ~41.4 |
| **Refinement** | | | | |
| Initial model used (PDB code) | 7UXH (mTOR, RAPTOR and RHEB are from AlphaFold) | 7UXH (mTOR and RHEB are from AlphaFold) | 7UXH (mTOR and RHEB are from AlphaFold) | 7UXH |
| Model resolution (Å) | 3.3 | 4.0 | 3.5 | 4.3 |
| FSC threshold | 0.5 | 0.5 | 0.5 | 0.5 |
| Map sharpening B factor (Å²) | −84.9 | −91.9 | −74.9 | −121.4 |
| Model composition | | | | |
| Non-hydrogen atoms | 80,794 | 22,106 | 22,420 | 11,246 |
| Protein residues | 10,098 | 2,754 | 2,792 | 1,421 |
| Ligands | 18 | 4 | 6 | 3 |
| B factors (Å²) | | | | |
| Protein | 83.03 | 141.97 | 92.39 | 108.48 |
| Ligand | 77.58 | 123.73 | 93.87 | 94.16 |
| R.m.s. deviations | | | | |
| Bond lengths (Å) | 0.004 | 0.004 | 0.004 | 0.005 |
| Bond angles (°) | 0.729 | 0.682 | 0.660 | 1.170 |
| **Validation** | | | | |
| MolProbity score | 1.80 | 1.83 | 1.67 | 1.92 |
| Clashscore | 7.99 | 9.61 | 7.22 | 8.94 |
| Poor rotamers (%) | 0.00 | 0.00 | 0.00 | 0.00 |
| Ramachandran plot | | | | |
| Favoured (%) | 94.75 | 95.35 | 96.03 | 93.05 |
| Allowed (%) | 5.13 | 4.61 | 3.97 | 6.74 |
| Disallowed (%) | 0.12 | 0.04 | 0.00 | 0.22 |

FSC, Fourier shell correlation.

intermediate and fully active states were 3.61 Å and 3.16 Å, respectively (Extended Data Fig. 1).

By aligning these states on the basis of the RHEB structure, we showed conformational changes between the intermediate and fully active states. The M-HEAT domain of the active state moved closer to the N-HEAT domain than it did with the intermediate state, with an average distance of about 5 Å (Fig. 4a and Extended Data Fig. 7a). The FAT and kinase domains of the fully active state moved about 5–7 Å towards RHEB compared with those in the intermediate state. As a result, the MLST8 subunit more than towards RHEB in the fully active state (Extended Data Fig. 7a). The soluble mTORC1–RHEB structure is in between the conformational pathway of the intermediate state and the fully active state when superimposed on the RHEB subunit (Extended Data Fig. 7b).

The mTOR–RHEB interaction is largely maintained in both intermediate and fully active states; however, the residues Met1255 and Lys1256 of the FAT domain of mTOR are disordered in the intermediate state but engaged with RHEB in the active state (Fig. 4b). In the mTOR intermediate state, Tyr35 in the switch I region of RHEB flips to the opposite side and is disengaged from the nucleotide, whereas, in the mTOR fully active state, the switch I region of RHEB adopts the canonical conformation of the RHEB–GTP-bound state. In this light, it is interesting that the mutation *RHEB*[Y35N] is linked to cancer[27] and expression of RHEB[Y35N] hyperactivates mTORC1 (refs. 28,29). Moreover, we observed an ordered loop in the fully active state, located in the opposite site of the adenosine triphosphate (ATP)-binding pocket, whereas the loop is disordered in the intermediate state and in the soluble mTORC1–RHEB structure (Extended Data Fig. 7c).

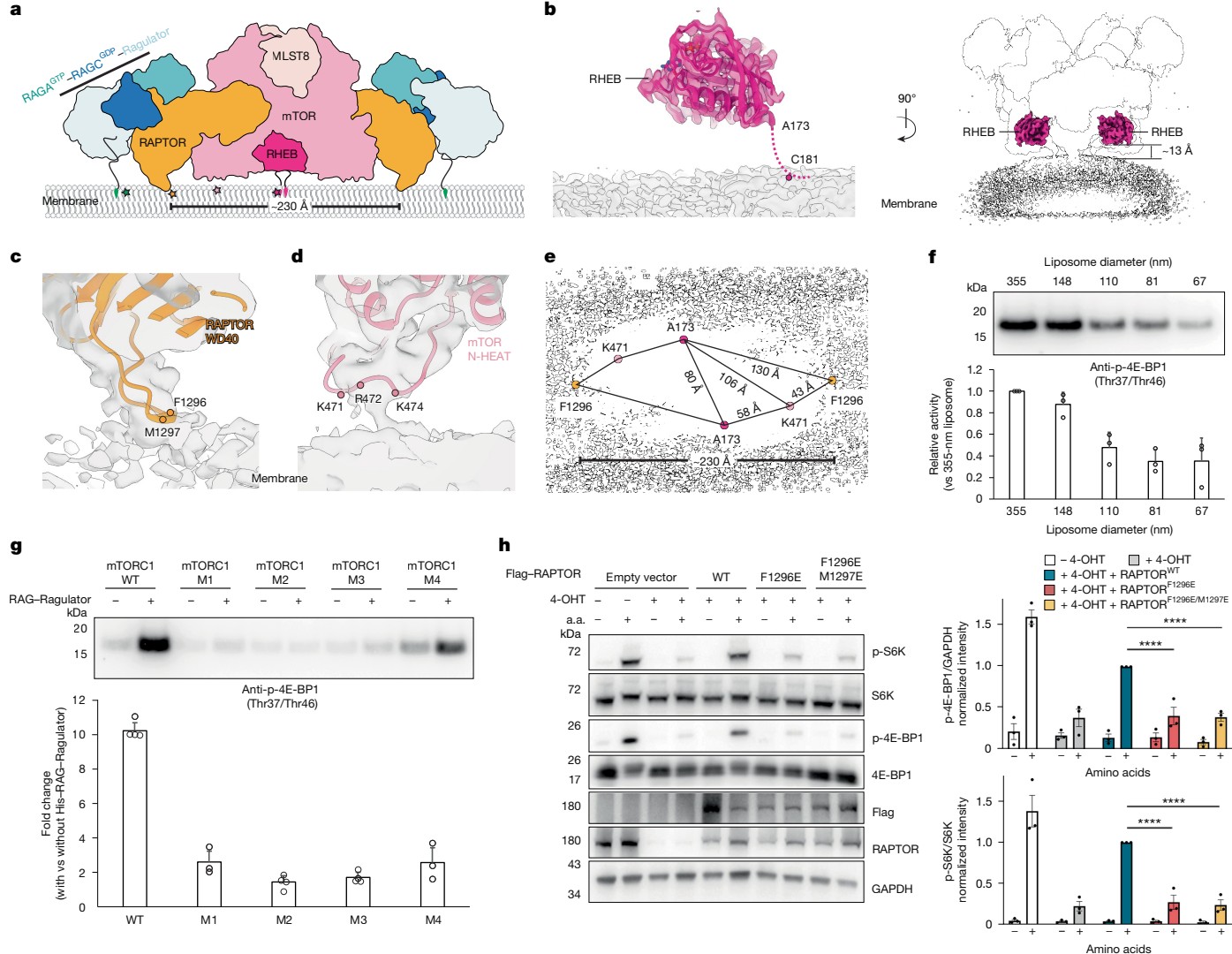

**Fig. 3 | Membrane-interacting sites of mTOR and RAPTOR subunits.**
**a**, Cartoon representation of the mTORC1–RHEB–RAG–Ragulator–4E-BP1 complex on a membrane shown from the side view. The distance between two RAPTOR membrane-interacting sites is shown. Coloured stars indicate the membrane contact sites in the asymmetric unit. The lipidation sites of LAMTOR1 and RHEB are indicated at the end of arbitrary linkers that anchor them to membranes. **b**, Close-up view of RHEB density (contour level, 0.168) above the membrane (left) and overall RHEB position relative to the membrane (right). The last visible residue is shown, and the potential membrane-tethering loop is indicated with a dashed line. Close-up view of membrane-interacting sites of the RAPTOR (**c**) and mTOR (**d**) subunits. For better visualization, the unsharpened cryo-EM map from the overall refinement with *C*2 symmetry is used (contour level, 0.033). Residues of RAPTOR and mTOR are indicated with dots. **e**, Geometry of the residues involved in membrane interaction. **f**, In vitro mTORC1 kinase activity with different liposome sizes; quantifications were calculated with three repeats and are mean ± s.d. **g**, In vitro kinase activity of mTORC1 mutants; quantifications were calculated with three (M1 and M4) or four (wild type (WT), M2 and M3) repeats and are mean ± s.d. mTORC1 M1, RAPTOR$^{F1296E/M1297E}$; mTORC1 M2, mTOR$^{468-476GS5}$ + RAPTOR$^{F1296E/M1297E}$; mTORC1 M3, mTOR$^{K471D/R472D/K474D}$ + RAPTOR$^{F1296E/M1297E}$; mTORC1 M4, mTOR$^{K471D/R472D/K474D}$. **h**, Inducible RAPTOR knockout MEFs, treated with 0.5 μM 4-hydroxytamoxifen (4-OHT) for 48 h or left untreated, were transfected with an empty vector control or constructs expressing RAPTOR$^{WT}$, RAPTOR$^{F1296E}$ or RAPTOR$^{F1296E/M1297E}$. After transfection (24 h), cells were starved of amino acids (a.a.) for 60 min or starved and restimulated with amino acids for 30 min and analysed by immunoblotting with the indicated antibodies. Quantifications of p-S6K/S6K and p-4E-BP1/GAPDH are shown with mean ± s.e.m. of *n* = 3 experiments (****$P$ < 0.0001, two-way ANOVA, Dunnett's multiple-comparison test). All samples derive from the same experiment, and gels and blots were processed in parallel.

To reveal local conformational differences in the kinase domain and in the ATP-binding pocket, we superimposed the intermediate and fully active states of mTOR on the basis of the C-lobe of the kinase domain. The FAT domain of mTOR is more engaged with the C-lobe in the fully active state than in the intermediate state (Fig. 4c and Extended Data Fig. 8a). The established electrostatic interactions between residues Glu1581 and Gln1355 in the FAT domain and between residues Arg1585 of the FAT domain and Glu2311 of the C-lobe in the fully active state indicate direct communication between the FAT domain and the kinase domain to promote the active state (Fig. 4c). In the ATP-binding pocket, the intermediate state had a conformation similar to that of the apo state, despite the drastic movement of the M-HEAT domain upon RHEB binding (Fig. 4d). In the fully active state, the ATP molecule was positioned 1.5 Å closer to the catalytic residues in the C-lobe than in the soluble mTORC1–RHEB structure, aligning more closely with other atypical serine–threonine kinases in the phosphatidylinositol-3-kinase-related kinase (PIKK) family in their active states (Extended Data Fig. 8b).

To characterize the relationships between different mTORC1 states and membrane engagement, we performed 3D variability analysis (3DVA) using the same symmetry-expanded and RAPTOR–RAG–Ragulator-subtracted particles that were used for 3D classification

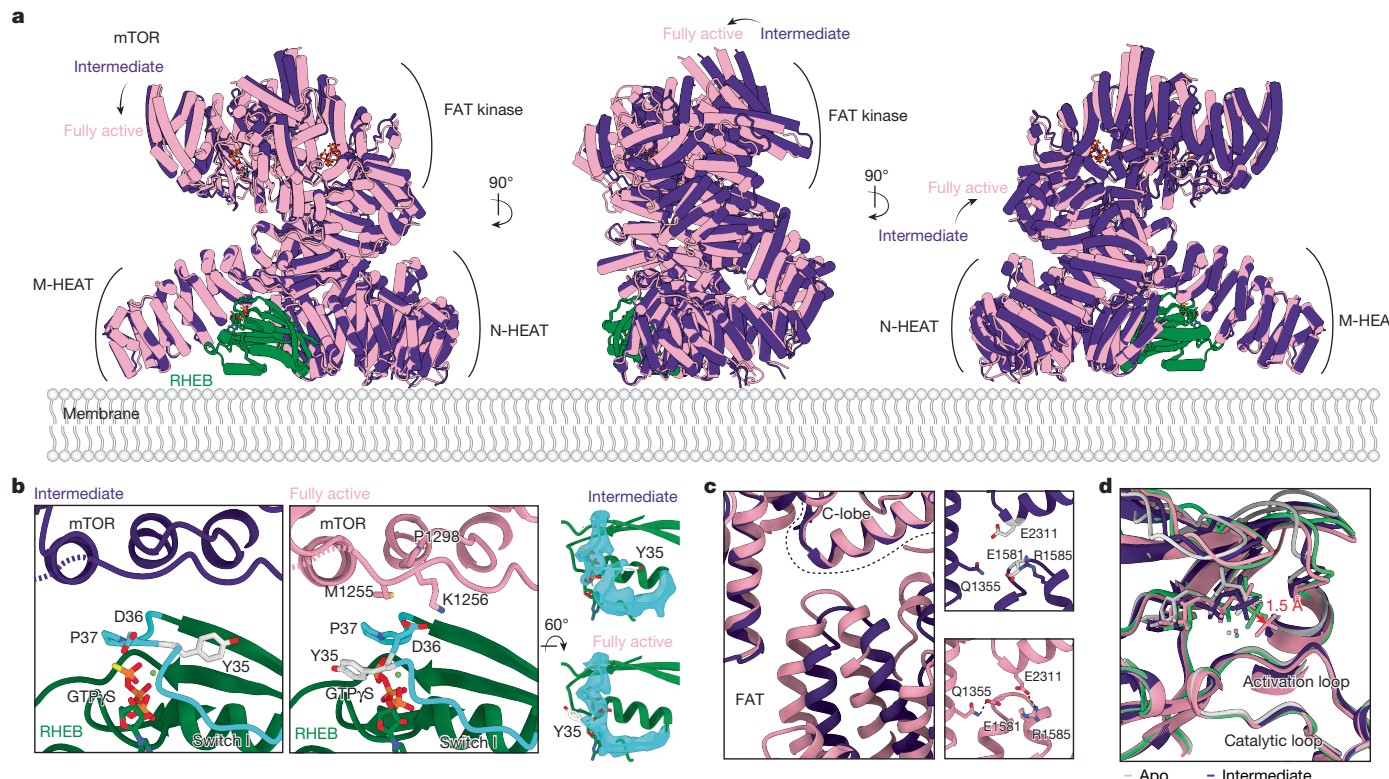

**Fig. 4 | Conformational flexibility of mTOR on the membrane. a**, The intermediate and fully active states of mTOR are superimposed by overlaying their bound RHEB molecules. Movement between intermediate and active states is indicated with arrows. **b**, Close-up view of the interaction between residues 1,255 and 1,260 of mTOR and switch I of RHEB in the intermediate and active states. The cryo-EM density corresponding to switch I is shown (contour levels of 0.139 and 0.137 for the intermediate state and the fully active state, respectively). **c**, Close-up view of interactions between the FAT and C-lobe domains of mTOR, with insets indicating residues in the intermediate and fully active states. The FAT (residues 1,255–1,453 are omitted) and kinase domains of the intermediate and active states are superimposed on the basis of the C-lobe (residues 2,200–2,400). **d**, Different states of mTOR structures are superimposed on the basis of the C-lobe (residues 2,200–2,400). The distance of the γ-phosphate of ATP between soluble mTORC1–RHEB and the fully active state is shown in red.

(Extended Data Fig. 9a). We identified one component that did not involve protein stretching into the membrane. We then extracted particles that were at the two ends of this variability component for further local refinement (Extended Data Fig. 9b). The two cryo-EM maps resembled the intermediate and fully active states identified in the 3D classification, as the absence or presence of the corresponding loop (residues 904–920) was confirmed, respectively (Extended Data Fig. 9c). The fully active state from 3DVA showed clear density for the basic loop of Lys471, Arg472 and Lys474 and an extended helix, indicating stronger membrane engagement, whereas the basic loop and the helix were not observed in the intermediate state (Fig. 5). The RAPTOR–membrane interaction was present in both intermediate and fully active states (Extended Data Fig. 9d). This implies that direct RAPTOR–membrane interaction is a general requirement for mTORC1 membrane engagement, whereas mTOR–membrane interaction promotes full activation of the mTOR kinase.

## Discussion

Among the questions we set out to address was why RHEB–GTP, which is critical for GF-dependent activation of mTORC1 in cells[3,5], is such a low-affinity activator in solution[4]. We found that physiological concentrations of lipidated RHEB–GTP, in the presence of liposomes, membrane-tethered Ragulator and an active RAG dimer, could potently activate mTORC1 in a biochemical reconstitution. This could be explained, at least in part, simply by the increased local concentration of RHEB–GTP in the vicinity of mTORC1. Reduction of dimensionality, which limits the search space accessible by diffusion to two dimensions[30], could be a contributing factor, but the limitations of this mechanism in biology have been noted[31]. We probed the mechanism more deeply using cryo-EM structure determination of the reconstituted system on the membrane. Although the addition of the membrane context greatly increases the realism of the system compared with previous membrane-free structural analyses, it is of course possible that yet further changes will be seen once it becomes feasible to reach atomistic resolution in situ and determine the structure as found on lysosomes in cells. We found that precise structural responses to the membrane context are also important.

By carrying out an atomically detailed analysis of intermediate and fully active conformations in the membrane context, we mapped an activation pathway mediated by the membrane itself. We found that both the mTOR and RAPTOR subunits of mTORC1 directly engage with the membrane. Membrane contacts by RAPTOR and mTOR residues separated by more than 230 Å promote large-scale conformational rearrangements of the N-HEAT, M-HEAT, FAT and kinase domains of mTOR. These large-scale changes in turn reorient the N-lobe and the C-lobe of the kinase domain to fine-tune the kinase active site in a catalytically optimal geometry. The contributions of membrane anchoring and membrane shape to catalytic activation were verified by reconstitution of membrane-binding site mutants of RAPTOR and mTOR and analysing the shape dependence of liposome activation. Lysosomes undergo tubulation and swelling in the course of their normal function and under stress[32,33], which raises the possibility that membrane shape changes could influence mTORC1 activation.

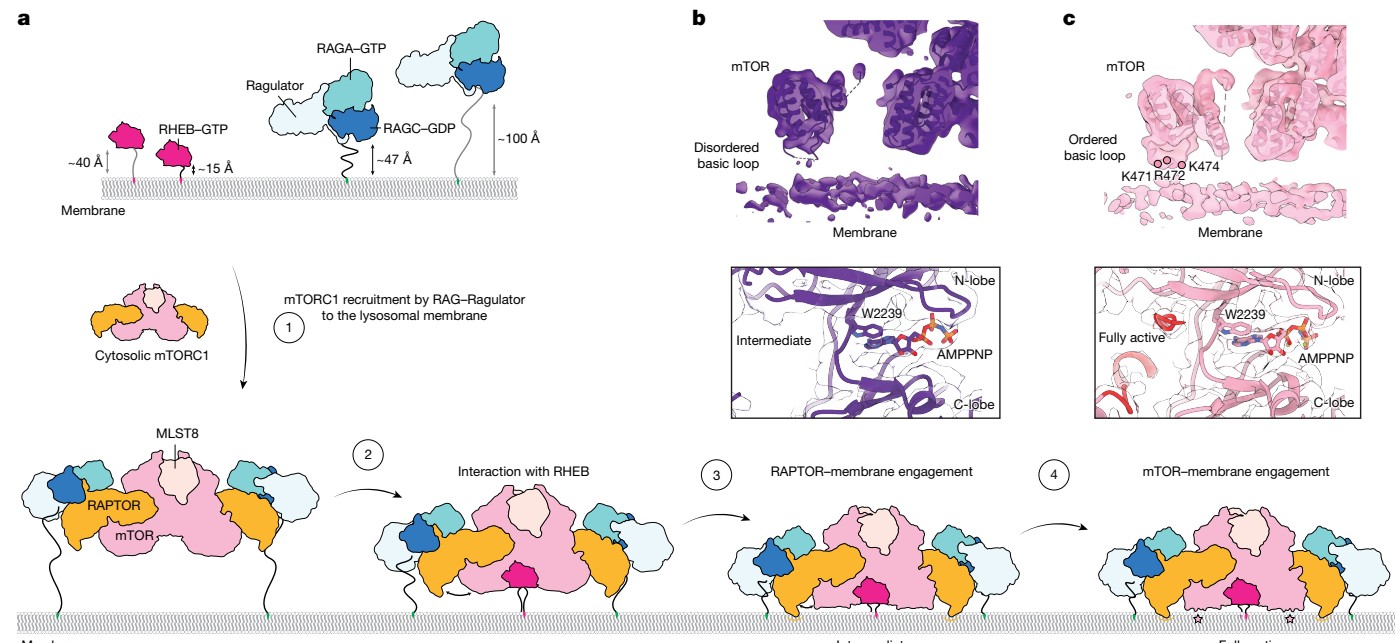

**Fig. 5 | A model of mTORC1 recruitment and activation on the lysosomal membrane. a**, Average distances between protein and membrane are indicated with black arrows. Grey arrows indicate the most possible extended positions relative to the membrane surface. Linkers that anchor LAMTOR1 and RHEB to membranes are arbitrary. Numbers indicate each step in the process of mTORC1 activation on the lysosomal membrane. Double-headed arrows indicate movement between RAPTOR and mTOR upon binding with RHEB. Curves and stars on the membrane indicate RAPTOR and mTOR engagement with the membrane, respectively. **b,c**, Top, close-up view of the cryo-EM density (from 3DVA, contour levels of 0.049 and 0.042 for the intermediate state and the fully active state, respectively) of the membrane-interacting site of mTOR in the intermediate (**b**) and fully active (**c**) states. Bottom, cryo-EM density (from 3D classification, contour levels of 0.13 and 0.11 for the intermediate state and the fully active state, respectively) of the ATP-binding pocket of the intermediate (**b**) and active (**c**) states.

mTOR is a member of the PIKK superfamily. Although the role of membrane anchors with large spatial separations in kinase activation is unique, similar local rearrangements of the N-lobes and C-lobes of other PIKK family members, ATM, MEC1[ATR] and DNA-PK, have been reported in response to reactive oxygen species-dependent activation[34], constitutive activating mutation (F2244L)[35] and DNA activation[36], respectively. We observed that the ATP molecule in the fully active state of membrane-bound mTORC1 aligns well with that of other activated PIKK family members, consistent with its identification as part of a more active conformation of mTORC1.

The observation in the membrane-bound structure of a second, previously unknown RAG–Ragulator binding site on MLST8 was unexpected. The MLST8 subunit is common to both mTORC1 and mTORC2, but the additional presence of SIN1 in mTORC2 blocks the previously unknown RAG–Ragulator site, consistent with the known lack of interaction of mTORC2 with RAGs. The previously unknown RAG–Ragulator site overlaps[4] with the binding site for a proline-rich AKT substrate of 40 kDa (PRAS40) on MLST8 in mTORC1. PRAS40 antagonizes GF-dependent mTORC1 activation[37–40]. MLST8 is dispensable for mTORC1 activity[41]. Therefore, it remains to be determined whether the previously unknown RAG–Ragulator site has a physiological activating role, whether by antagonizing PRAS40 inhibition or simply by augmenting lysosomal recruitment. Another open question concerns the role of membrane contacts in mTORC1 phosphorylation of TFEB and other noncanonical substrates, which requires RAG–Ragulator but not RHEB[6,20,42].

mTORC1 is activated on lysosomes in response to GF signalling, despite the fact that only a small fraction of the key mediator of GF signalling, RHEB, is transiently localized on lysosomes[43,44]. Physiologically, this serves to AND gate the signal to proliferate with the availability of amino acids needed as building blocks for growth. Yet it has been unclear how signal integration is executed at the structural levels.

The RAG GTPase dimer is tethered to the lysosome by the Ragulator complex. Ragulator is anchored by lipidation of its LAMTOR1 subunit, and the lipid anchor is connected to the folded core of Ragulator by a 45-residue N-terminal disordered region[23,24,45–47], with an estimated end-to-end length of ~47 Å on average and an extended length[48] of ~100 Å (Extended Data Fig. 10a). The distance is potentially longer because of volume exclusion effects owing to its tethering to the membrane surface[49]. Augmented to the dimensions of the RAG–Ragulator complex leads us to estimate that the membrane-docking site of mTOR would be tethered within ~40 Å of the membrane surface upon RAG–Ragulator engagement. This is close enough to markedly increase the probability of encountering membrane-tethered RHEB, which itself is tethered at a mean distance of ~15 Å and an extended distance of ~40 Å from the membrane (Extended Data Fig. 10b). Even RHEB is still not close enough to drive full membrane engagement and activation on its own. Thus, lysosomal membrane engagement and enzyme activation is a four-step process, in which (1) initial localization to within an approximate vicinity of 10 nm of the membrane is driven by RAG–Ragulator. This allows (2) capture by RHEB at a distance of ~1.5–4 nm from the membrane. At this stage, the large conformational change characteristic of the RHEB-bound state[4] occurs. Close docking to the membrane is driven in the first instance (3) by the RAPTOR FM finger, which contacts the membrane in both the intermediate and fully active states. Finally, the mTOR–membrane-interacting site is only seen to dock in the fully active state (4), suggesting that this interaction is involved in finally reaching the highest level of activation (Fig. 5). One remaining challenge is relating these structural transitions in a precise way to changes in catalytic rates. Further kinetic investigation of this model using single-molecule Förster resonance energy transfer probes might further advance understanding of the relationship between enzyme kinetics on the one hand and movements at various size scales on the other hand. This model provides a satisfying structural explanation

for how mTORC1 integrates GF and nutrient signals in the context of the lysosomal membrane.

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

## Methods

### Protein expression and purification

The full-length, codon-optimized genes encoding human RAGC with the S75N substitution and human RAGA with the Q66L substitution were synthesized (Twist Bioscience) and individually cloned into a pCAG vector. Mutations were introduced in the genes encoding RAGA (Q66L) and RAGC (S75N) to lock the active state of RAG GTPases (RAGA[GTP]–RAGC[GDP]). The RAGC (S75N) construct included sequence for a TEV-cleavable GST tag at the N terminus, whereas the RAGA (Q66L) construct was tagless. For expression and purification of RAG GTPases, HEK293F GnTI- cells were transfected with a total of 1 mg plasmid DNA (550 µg for RAGA and 450 µg for RAGC) and 4 mg polyethylenimine (PEI) (Sigma-Aldrich) per l at a density of $1.8 \times 10^6$ cells per ml. Cells were collected after 72 h and lysed by gentle nutation in wash buffer (50 mM HEPES, 150 mM NaCl, 2.5 mM $MgCl_2$, 1 mM TCEP, pH 7.4) supplemented with 0.4% CHAPS and Protease Inhibitor (Roche) for 1 h. The lysate was cleared by centrifugation at 35,000$g$ for 35 min. The supernatant was incubated with glutathione Sepharose 4B (GE Healthcare) resin for 2 h. The resin was then washed first with a modified wash buffer (200 mM NaCl and 0.3% CHAPS) and then with wash buffer. The complex was eluted by on-column TEV cleavage overnight without nutation. Eluted complexes were concentrated and further purified by size exclusion chromatography (SEC) using a Superose 200 10/300 GL (GE Healthcare) column equilibrated with wash buffer.

The full-length, codon-optimized genes encoding the human Regulator complex (*LAMTOR1–LAMTOR5*) were synthesized (Twist Bioscience) and individually cloned into a pCAG vector. The Lamtor1 construct includes sequence for a TEV-cleavable GST tag at the N terminus, followed by sequence for a His[6] tag, which replaces its first seven residues. The Lamtor2 construct features sequence for a TEV-cleavable tandem 2× Strep II-1× Flag tag (2SF-TEV) at the N terminus. A total of 1 mg plasmid DNA (200 µg each of the five constructs) and 4 mg PEI (Sigma-Aldrich) were used to transfect HEK293F GnTI- cells at a density of $1.8 \times 10^6$ cells per ml per l. The cells were pelleted after 72 h of transfection and lysed in wash buffer containing 1% Triton X-100 and protease inhibitor. The cleared supernatant after centrifugation was applied to glutathione Sepharose 4B (GE Healthcare) resin and incubated for 2 h. The complex was eluted by on-column TEV cleavage overnight without nutation and supplemented with a final concentration of 0.5 mM EDTA. Further purification was performed by SEC using a Superdex 200 10/300 GL column equilibrated with wash buffer. The fractions containing all five subunits were pooled and concentrated.

To assemble the RAG–Regulator complex, an excess amount of RAG was incubated with Regulator and Ni-NTA resin (Thermo Scientific) at 4 °C for 1 h. Excess RAG was removed by washing the resin with wash buffer containing 40 mM imidazole. The assembled RAG–Regulator complex was eluted using wash buffer with 250 mM imidazole and further purified by SEC with a Superdex 200 10/300 GL column.

The human mTORC1 complex was purified in a manner similar to that of previous methods[20]. The full-length, codon-optimized genes encoding human mTOR, RAPTOR and MLST8 were cloned into pCAG vectors (mTOR with TEV-cleavable 2SF, RAPTOR with uncleavable 2SF and MLST8 with uncleavable 2SF). A total of 1.35 mg plasmid DNA (900 µg for mTOR, 250 µg for RAPTOR and 200 µg for MLST8) and 4 mg PEI (Sigma-Aldrich) were used to transfect HEK293F GnTI- cells at a density of $1.8 \times 10^6$ cells per ml per l. Cells were collected 72 h after transfection and lysed in the same buffer used for RAGA–RAGC purification. The complex was purified using Strep-Tactin resin (IBA Lifesciences) and eluted with wash buffer containing 10 mM D-desthiobiotin. The eluate was then diluted in an equal volume of salt-free buffer (50 mM HEPES, 1 mM TCEP, pH 7.4) and applied to a 1-ml HiTrap Q column (GE Healthcare). The mTORC1 complex and free RAPTOR were separated with a 20-ml salt gradient to a final concentration of 0.5 M NaCl, using salt-free buffer and high-salt buffer (50 mM HEPES, 1 M NaCl, 1 mM TCEP, pH 7.4). The flow rate was 0.2 ml min$^{-1}$, and the fraction size was 0.2 ml. The fractions containing the mTORC1 complex and free RAPTOR were collected and concentrated with Amicon Ultra-4 concentrators. Mutations in the genes encoding mTOR and Raptor were generated using NEBuilder HiFi DNA Assembly. The mTORC1 complex mutants were produced by altering the combination of wild-type and mutated genes during transfection and purified as described above.

Plasmids containing full-length genes encoding human 4E-BP1 and RHEB were gifts from the Zoncu laboratory (University of California, Berkeley). These genes were individually cloned into a 2GT vector from QB3 MacroLab (https://qb3.berkeley.edu/facility/qb3-macrolab/), which features sequence for a TEV-cleavable tandem GST–His[6] tag at the N terminus. Genes encoding 4E-BP1 and RHEB were overexpressed in *Escherichia coli* Rosetta 2(DE3) strains and purified using the same method. *E. coli* cells were grown in LB medium at 37 °C until an OD of 0.6 was reached and then induced by adding 0.2 mM IPTG at 18 °C overnight. Cells were collected, resuspended in Ni buffer (50 mM Tris-Cl, pH 7.5, 300 mM NaCl, 20 mM imidazole, 5 mM 2-mercaptoethanol, 1 mM PMSF) and lysed by sonication. Protein was purified using HisPur Ni-NTA Resin (Thermo Scientific), washed with Ni buffer containing 40 mM imidazole and eluted with 250 mM imidazole. The GST–His[6] tag was cleaved by incubating with TEV enzyme overnight. Further purification was performed by SEC using a Superdex 75 10/300 GL column equilibrated with buffer (50 mM HEPES, pH 7.5, 150 mM NaCl, 0.5 mM TCEP). Fractions containing the desired proteins were passed through the Ni column to remove residual GST–His[6] and then concentrated.

To charge RHEB with GTPγS (Abcam) or GDP (Sigma-Aldrich), RHEB was first diluted in buffer (50 mM HEPES, pH 7.5, 150 mM NaCl, 1 mM TCEP, 5 mM EDTA) and then supplemented with GDP or GTPγS at a 30-fold molar excess. The mixture was incubated at 30 °C for 1 h, followed by adding 20 mM $MgCl_2$. Further purification was carried out by SEC using a Superose 75 10/300 GL (GE Healthcare) column to remove excess nucleotides.

All steps of protein purification were performed at 4 °C, and aliquoted proteins were flash frozen in liquid nitrogen and stored at −80 °C.

### LUV preparation

A lipid mixture was prepared in a glass vial using the lipid composition shown in Supplementary Table 1. To form a thin film on the glass wall, the glass vial was slowly shaken on a vortex while drying under nitrogen gas. The glass vial was then placed in a vacuum oven overnight at room temperature to evaporate any remaining solvent. Lipids were hydrated in a lipid buffer (25 mM HEPES, pH 7.2, 100 mM NaCl) to a final concentration of 1.8 mM for 1 h, with intermittent vortexing during hydration. The solution was transferred to a 15-ml Eppendorf tube and subjected to nine freeze–thaw cycles using liquid nitrogen and a 40 °C water bath. The lipid mixture was either stored at −80 °C or immediately extruded using an Avanti Polar Lipids Mini Extruder (610023) at least 41 times through a 200-nm filter (Whatman Nuclepore Track-Etched Membranes, diameter of 19 mm) for mTORC1 kinase activity assays and cryo-EM studies. Filters of different diameters were also used to generate liposomes of various sizes to assess the effect of liposome size on mTORC1 kinase activity. The mean diameter of extruded liposomes was measured using dynamic light scattering (Zetasizer Ultra). The average sizes of liposomes for filters of 400 nm, 200 nm, 100 nm, 50 nm and 30 nm were about 355 nm, 148 nm, 110 nm, 81 nm and 67 nm, respectively. The lipid solution after extrusion was stored at 4 °C for up to 2 weeks.

### mTORC1 kinase activity with LUVs

The kinase assay was conducted in a buffer containing 25 mM HEPES (pH 7.2), 100 mM NaCl, 10 mM imidazole, 10 mM $MgCl_2$ and 2 mM DTT at 30 °C for 10 min in a final volume of 50 µl. First, RHEB was incubated with liposomes at concentrations of 0.25 µM and 0.18 mM, respectively, at 4 °C overnight in 40 µl lipid buffer. In parallel, RHEB alone

and liposomes alone were diluted to the same concentrations in lipid buffer and incubated at 4 °C overnight. After overnight incubation, the reactions were stopped with 2 mM DTT at room temperature for 30 min, followed by adding 10 mM $MgCl_2$ and 10 mM imidazole. The $His_6$-tagged RAG–Ragulator complex was then added to the reactions and incubated on ice for 30 min. Subsequently, 4E-BP1 and mTORC1 were added to the reactions at final concentrations of 10 μM and 5 nM, respectively. The reactions were initiated by adding ATP at a final concentration of 1 mM and incubated in a thermocycler (Bio-Rad, T100) at 30 °C for 10 min. The reactions were stopped by diluting tenfold into a urea denaturing buffer (50 mM Tris-Cl, pH 7.5, 150 mM NaCl, 8 M urea). All reactions were then diluted into 4× NuPAGE LDS sample buffer and boiled for 2 min. Proteins were resolved on a 4–12% NuPAGE Bis-Tris gel and transferred to PVDF membranes using the Trans-Blot Turbo Transfer System. Western blotting was performed using the anti-p-4E-BP1 antibody (Cell Signaling Technology, 2855, 1:10,000 dilution). For dot blot analysis, 2 μl of the denatured sample in urea buffer was directly applied to nitrocellulose membranes, and protein was detected with the same 4E-BP1 antibody. Unprocessed images are included in Supplementary Fig. 1.

### Cryo-EM sample preparation and imaging

The mTORC1–RHEB–RAG–Ragulator–4E-BP1 complex on liposomes was reconstituted with the following steps. First, a mixture of RHEB and liposomes (200 nm) was incubated at 4 °C overnight in lipid buffer at concentrations of 8 μM and 1.8 mM, respectively. The RHEB–liposome mixture was supplemented with 2 mM DTT and 10 mM $MgCl_2$ and incubated at room temperature for 30 min. In parallel, mTORC1 was incubated with $His_6$–RAG–Ragulator in lipid buffer containing 5 mM TCEP at concentrations of 1 μM and 4 μM, respectively. Equal volumes of the RHEB–liposome mixture and the mTORC1–RAG–Ragulator mixture were then combined and incubated on ice for 2 h. Finally, 4 μM of 4E-BP1 and 1 mM of AMPPNP were added to the mixture for 10 min before application to cryo-EM grids for vitrification.

Cryo-EM samples were prepared by applying 3 μl of the aforementioned complex to a glow-discharged (PELCO easiGlow, 45 s in air at 15 mA and 0.37 mbar) holey carbon grid (C-flat, 2/1-3C-T) and vitrified using the FEI Vitrobot Mark IV System (Thermo Fisher Scientific). The samples were incubated on grids for 1 min and blotted for 3 s with a blot force of 15, using two Whatman 595 papers on the sample side and one Whatman 595 paper on the backside, at 6 °C with 95% relative humidity.

Cryo-EM images of the mTORC1–RHEB–RAG–Ragulator–4E-BP1 complex on liposomes were recorded using a Titan Krios G3 microscope (Thermo Fisher Scientific) equipped with a Gatan Quantum energy filter (slit width of 20 eV) and operated at 300 kV. Automated data acquisition was performed using SerialEM[50] on a K3 Summit direct detection camera (Gatan) in superresolution correlated double-sampling mode with a pixel size of 0.52 Å and a defocus range of −0.9 to −2.2 μm. A total of 36 exposures per stage shift were enabled by large beam shift. Beam intensity was adjusted to a dose rate of around 1 e$^-$ Å$^{-2}$ per frame for a 30-frame movie stack with a total exposure time of 5.4 s. A total of 58,092 movies were recorded.

### Cryo-EM data processing

The data-processing scheme for the mTORC1–RHEB–RAG–Ragulator–4E-BP1 complex on liposomes using cryoSPARC (v.4)[51] is shown in Extended Data Fig. 1, and statistics are summarized in Table 1. Owing to the uneven distribution of liposomes in grid squares, 29,301 micrographs containing liposomes were manually selected for processing. Owing to the size of the dataset, micrographs were split and processed following the same protocol and then combined at the homogeneous refinement stage. Initially, the Blob Picker was used to maximize the number of particles. Two-dimensional classification was only used to remove obvious junk particles (for example, ice and chaperonin contaminants). An initial model of mTORC1 from an ab initio reconstruction

in a previous dataset and three bad initial models from ab initio reconstruction in this dataset were used for heterogeneous refinement. Iterative heterogeneous refinement was performed to select good particles, retaining rare views that may not have been identified in 2D classification. To minimize the possibility of membrane density biasing the alignment in the first iterations of refinement, the original initial models were used instead of reconstructions from each heterogeneous refinement. After extensive cleaning using heterogeneous refinement, particles were merged, and duplicates were removed with a cutoff distance of 50 Å. Additional heterogeneous refinement was used to further sort out particles. Homogeneous refinement was then performed for the full dataset. After identifying a good particle set, it was used to train Topaz particle picking[52]. Particles from Topaz picking underwent the same sorting procedure and were merged with the blob-picked particles, with duplicates removed using a cutoff distance of 50 Å.

In total, 337,347 particles containing protein complexes and membranes were selected, and reference-based motion correction was used to produce polished particles[53]. Symmetry expansion, particle subtraction and local refinement were used to produce an overall cryo-EM map of an asymmetric unit. Focused 3D classification was then used to separate particles without extra RAG–Ragulator copies and with either one or two extra copies of RAG–Ragulator. Further particle subtraction and local refinement were used to focus on either mTOR–RHEB–MLST8 or RAPTOR–RAG–Ragulator subcomplexes. Focused 3D classification was used to separate the intermediate and fully active states of mTOR. Populations containing extra copies of RAG–Ragulator were pooled together, and further particle subtraction and local refinement were used to obtain the cryo-EM map of MLST8–RAG–Ragulator.

In summary, 179,506 particles were refined to 3.23 Å with $C2$ symmetry for the mTORC1–RHEB–RAG–Ragulator–4E-BP1 complex without extra RAG–Ragulator copies on the membrane. Local refinement of mTOR–RHEB–MLST8 (359,012 particles after symmetry expansion) and RAPTOR–RAG–Ragulator (189,975 particles after symmetry expansion) yielded maps of 3.12 Å and 2.98 Å, respectively. Further classification of mTOR–RHEB–MLST8 identified the fully active and intermediate conformations. Final refinement of mTOR–RHEB–MLST8 in fully active (133,193 particles after symmetry expansion) and intermediate (109,105 particles after symmetry expansion) conformations resulted in resolutions of 3.16 Å and 3.61 Å, respectively. Final resolutions for the mTORC1–RHEB–RAG–Ragulator–4E-BP1 complex containing either one (128,189 particles) or two (29,652 particles) extra copies of RAG–Ragulator were 3.47 Å ($C1$ symmetry) and 3.81 Å ($C2$ symmetry), respectively. Three-dimensional variability analysis for populations without extra RAG–Ragulator copies was accomplished with cryoSPARC v.4 (Extended Data Fig. 9).

The overall resolution of all these reconstructed maps was assessed using the gold-standard criterion of Fourier shell correlation[54] at a cutoff of 0.143 (ref. 55). cryoSPARC v.4 was used to estimate local resolution.

### Atomic model building and refinement

To build the atomic model for the mTORC1–RHEB–RAG–Ragulator–4E-BP1 complex on the membrane, we first fit our previous models into the cryo-EM map as a rigid body using UCSF ChimeraX[56], with substituted models of mTOR and RAPTOR from AlphaFold2 prediction[57]. A composite map combining the local refinement maps was assembled in UCSF ChimeraX. Model refinement against local maps was accomplished using PHENIX for real-space refinement[58]. Manual model building was conducted with Coot[59] and ISOLDE[60] to iteratively inspect and improve local fitting. All figures were created using UCSF ChimeraX.

### Cell culture

Inducible RAPTOR knockout MEFs were kindly provided by M.N. Hall (University of Basel). MEFs were cultured in DMEM High Glucose medium (ECM0728L, Euroclone) supplemented with 10% inactivated

FBS (ECS0180L, Euroclone), 2 mM glutamine (ECB3000D, Euroclone), penicillin (100 IU ml⁻¹) and streptomycin (100 µg ml⁻¹) (ECB3001D, Euroclone) and maintained at 37 °C with 5% $CO_2$. All RAPTOR mutants used in these cellular assays were generated using the QuikChange II-E Site-Directed Mutagenesis Kit (200555, Agilent Technologies). Cells were transfected in 10-cm dishes using Lipofectamine 2000 Transfection Reagent (Invitrogen).

## Cell treatment

For experiments involving amino acid starvation, cells were rinsed twice with PBS and incubated for 60 min in amino acid-free DMEM (MBS6120661) supplemented with 10% dialysed FBS. Serum was dialysed against 1× PBS through dialysis tubing (molecular weight cutoff of 3,500 Da) to ensure the absence of contaminating amino acids. For amino acid refeeding, cells were restimulated for 30 min with a 1× water-solubilized mix of essential (11130036, Thermo Fisher Scientific) and non-essential (11140035, Thermo Fisher Scientific) amino acids resuspended in amino-acid-free DMEM supplemented with 10% dialysed FBS plus glutamine.

## Western blotting

Antibodies used in cellular studies include anti-p-p70 S6K (Thr389) (1A5) (mouse mAb, 9206, 1:1,000 for western blotting), anti-p70 S6K (rabbit, 9202, 1:1,000 for western blotting), anti-4E-BP1 (rabbit, 9644, 1:1,000 for western blotting), anti-p-4E-BP1 (Ser65) (rabbit, 9456, 1:1,000 for western blotting) and anti-RAPTOR (24C12) (rabbit, 2280, 1:1,000 for western blotting) from Cell Signaling Technology; anti-GAPDH (6C5) (rabbit, sc-32233, 1:15,000 for western blotting) from Santa Cruz; and anti-Flag M2 (mouse, F1804, 1:1,000 for western blotting) from Sigma-Aldrich. Cells were rinsed once with PBS and lysed in ice-cold lysis buffer (250 mM NaCl, 1% Triton, 25 mM HEPES, pH 7.4) supplemented with protease and phosphatase inhibitors. Total lysates were passed ten times through a 25-gauge needle with a syringe, kept at 4 °C for 10 min and then cleared by centrifugation in a microcentrifuge (14,000 rpm at 4 °C for 10 min). Protein concentration was measured with the Bradford assay. We performed densitometry analysis to calculate the intensity of phosphorylated and total protein using ImageJ software. The ratios between the values of phosphorylated and total protein were normalized to those of a control condition. Values in quantitative graphs are mean ± s.e.m. of at least three independent experiments. For statistical analysis, two-way ANOVA and Dunnett's post hoc test were used to compare differences between groups that had been split into two factors. Unprocessed gel images are shown in Supplementary Fig. 1.

## Reporting summary

Further information on research design is available in the Nature Portfolio Reporting Summary linked to this article.

## Data availability

Structural coordinates were deposited in the PDB with accession codes PDB 9ED4 (mTORC1–RAG–Ragulator–4E-BP1), PDB 9ED6 (MLST8–RAG–Ragulator), PDB 9ED7 (fully active state of mTOR–RHEB) and PDB 9ED8 (intermediate state of mTOR–RHEB). Cryo-EM density maps were deposited in the Electron Microscopy Data Bank with accession numbers EMD-47932 (a composite map of mTORC1–RAG–Ragulator–4E-BP1 on the membrane), EMD-47933 (focused refinement of the MLST8–RAG–Ragulator subcomplex), EMD-47934 (mTORC1–RAG–Ragulator–4E-BP1 on the membrane with two extra RAG–Ragulator copies), EMD-47935 (mTORC1–RAG–Ragulator–4E-BP1 on the membrane with one extra RAG–Ragulator copy), EMD-47936 (mTORC1–RAG–Ragulator–4E-BP1 complex on the membrane with *C*2 symmetry), EMD-47937 (focused refinement of mTORC1–RAG–Ragulator–4E-BP1 on the membrane with mTOR–MLST8–RHEB mask), EMD-47938 (focused refinement of mTORC1–RAG–Ragulator–4E-BP1 on the membrane with Raptor–RAG–Ragulator mask), EMD-47939 (fully active state of mTOR on the membrane) and EMD-47940 (intermediate state of mTOR on the membrane). All other data are provided in the Article and its Supplementary Information. Materials are available by request from the corresponding author with no restrictions beyond those of the Uniform Biological Material Transfer Agreement. Source data are provided with this paper.

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

**Acknowledgements** We thank D. Toso and R. Thakkar for cryo-EM facility support and members of the Hurley laboratory for insightful discussions. This work was supported by Genentech as part of the Alliance for Therapies in Neuroscience and the National Cancer Institute (NIH, R01 CA285366) (J.H.H.); the Italian Telethon Foundation (to G.N. and A.B.); the Fondazione AIRC per la Ricerca sul Cancro (MFAG-23538 to G.N. and IG-29230 to A.B.); MIUR (PRIN 2022CRFNCP to G.N., PRIN 202032AZT3 to A.B. and PRIN P2022T4PKT to G.N. and A.B.); Wereld Kanker Onderzoek Fonds, as part of the World Cancer Research Fund International grant programme (IIG_FULL_2022_009 to G.N.); and the European Research Council (INCANTAR, grant 101097752 to A.B.).

**Author contributions** Z.C. and J.H.H. conceived and designed research, Z.C. and A.E. carried out research, G.N., A.B. and J.H.H. supervised research, Z.C. and J.H.H. wrote the first draft, and all authors edited the manuscript.

**Competing interests** J.H.H. is a cofounder and shareholder of Casma Therapeutics and receives research funding from Hoffmann-La Roche. A.B. is a cofounder and shareholder of Casma Therapeutics and an advisory board member of Avilar Therapeutics and Amplify Therapeutics. G.N. is an advisory board member of Amplify Therapeutics. The other authors declare no competing interests.

**Additional information**
**Correspondence and requests for materials** should be addressed to James H. Hurley.

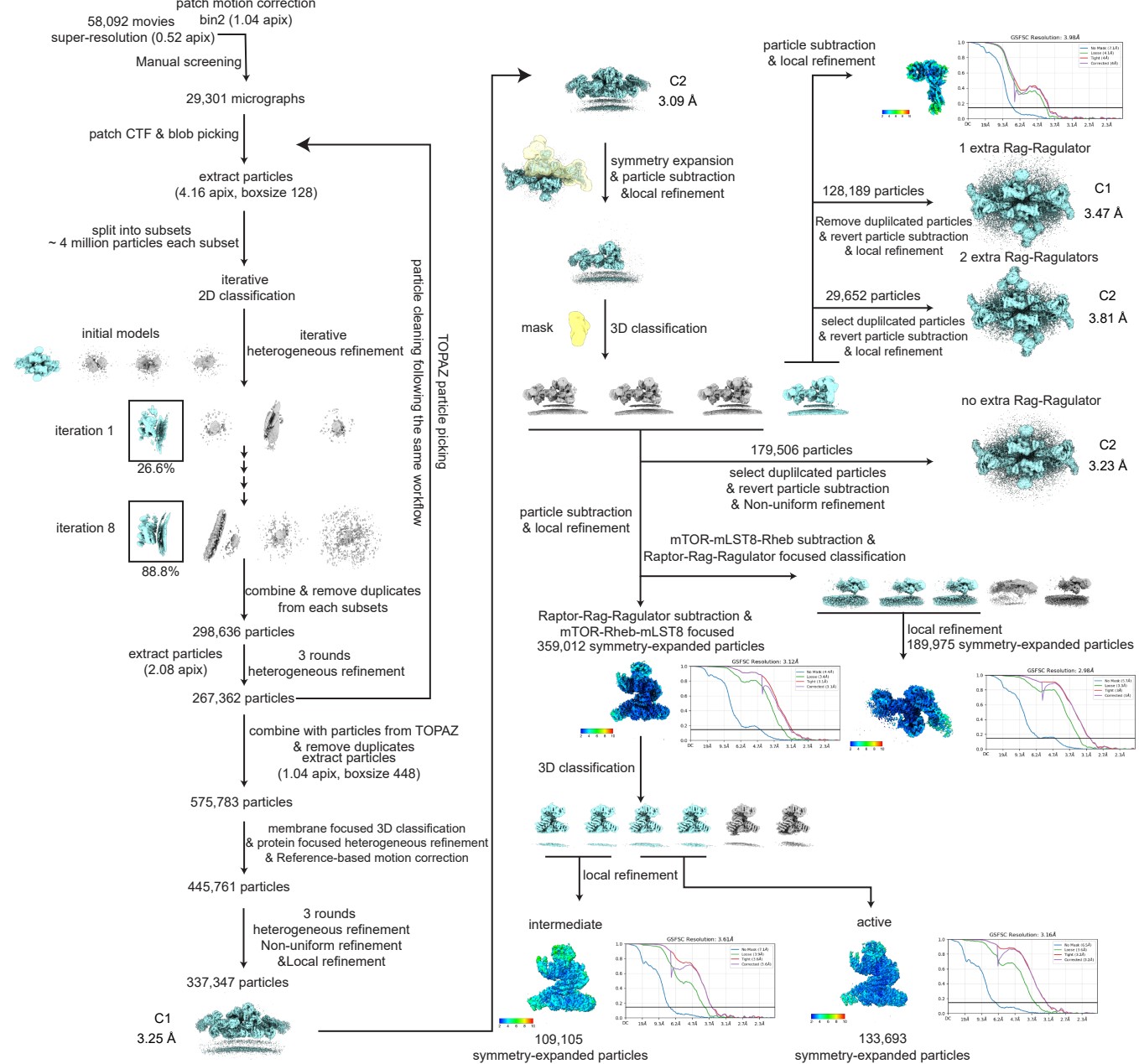

**Extended Data Fig. 1 | Cryo-EM workflow of mTORC1-Rheb-Rag-Ragulator-4EBP1 on membrane.** Intermediate cryo-EM density maps of iterative heterogenous refinement are shown, particles belonging to the boxed maps are selected for the next round of heterogeneous refinement. The percentage of the particles in the good class in each iteration is indicated.

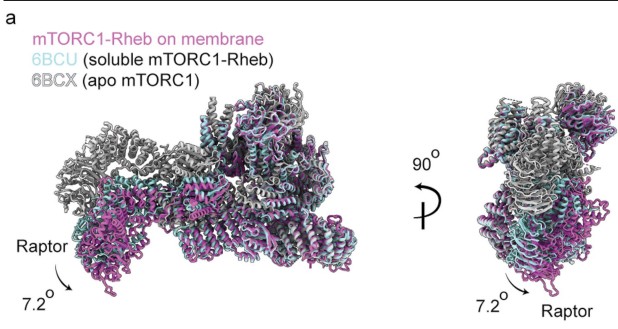

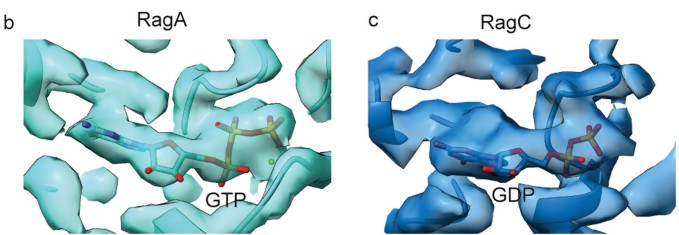

**Extended Data Fig. 2 | Comparison between mTORC1-Rheb on membrane and in solution and Representative cryo-EM density of the active sites of Rag GTPases. a**, Asymmetric units of apo mTORC1, mTORC1-Rheb in solution, and mTORC1-Rheb on membrane are aligned based on the mTOR subunit. The rotational direction of Raptor subunit between soluble and membrane-bound mTORC1-Rheb is indicated with an arrow. Cryo-EM density of RagA (**b**) and RagC (**c**) at contour level 0.25 and level 0.2, respectively.

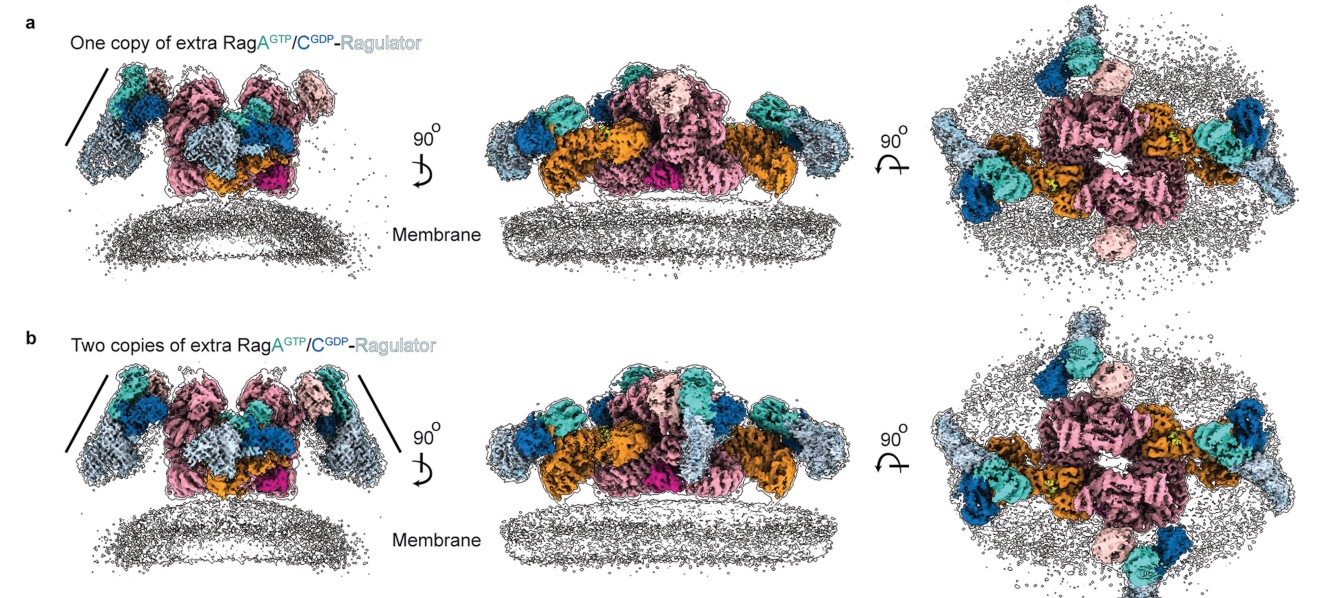

**Extended Data Fig. 3 | Cryo-EM maps of mTORC1-Rheb-Rag-Ragulator-4EBP1 with extra copy of Rag-Ragulator. a**, Cryo-EM density map of mTORC1-Rheb-Rag-Ragulator-4EBP1 on membrane with one extra copy of Rag-Ragulator, overlaid with the unsharpened map from the overall refinement with C1 symmetry. **b**, Cryo-EM density map of mTORC1-Rheb-Rag-Ragulator-4EBP1 on membrane with one extra copy of Rag-Ragulator, overlaid with the unsharpened map from the overall refinement with C2 symmetry.

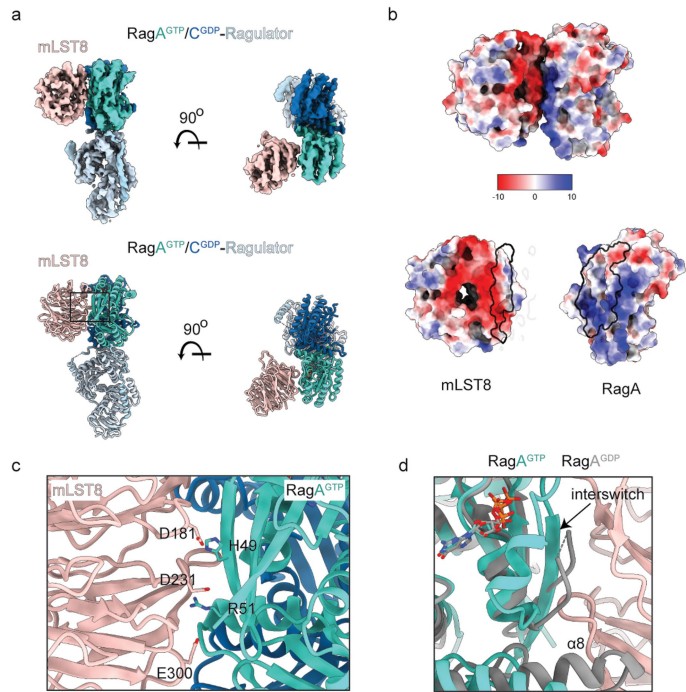

**Extended Data Fig. 4 | The interaction between mLST8 and RagA.**
**a**, Cryo-EM density map (contour level 0.149) and model of mLST8-Rag-Ragulator subcomplex. **b**, Electrostatic surface potentials of mLST8 and RagA are shown. The interaction surfaces are outlined. **c**, Close-up view of the interaction between mLST8 and RagA. **d**, Overlay between GTP-bound and GDP-bound RagA (PDB: 6NZD), the missing interswitch region is indicated with arrow. The GDP-bound RagA has a disordered interswitch region, which is responsible for the interaction with mLST8. In addition, the α8 helix in the RagA-GDP has potential clash with mLST8.

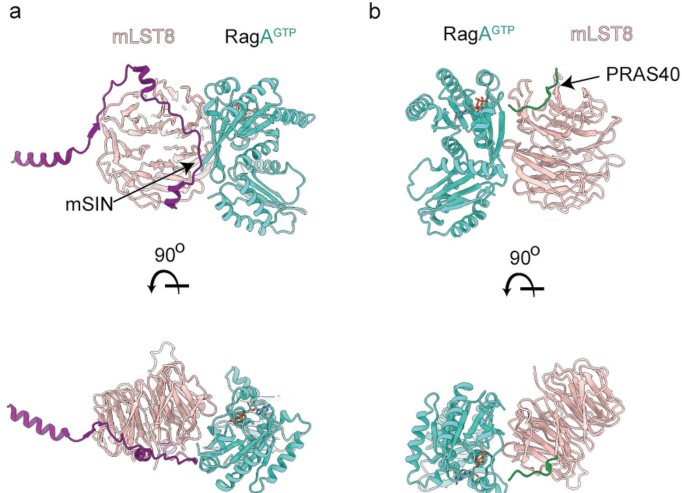

**Extended Data Fig. 5 | Relative position of mSIN and PRAS40 to mLST8-bound RagA.** Structures are superimposed based on mLST8. **a**, A potential clash is observed between mSIN and RagA, indicating this interaction is not compatible with mTORC2. **b**, The mLST8-interacting fragment of PRAS40 is localized close to the mLST8-bound RagA.

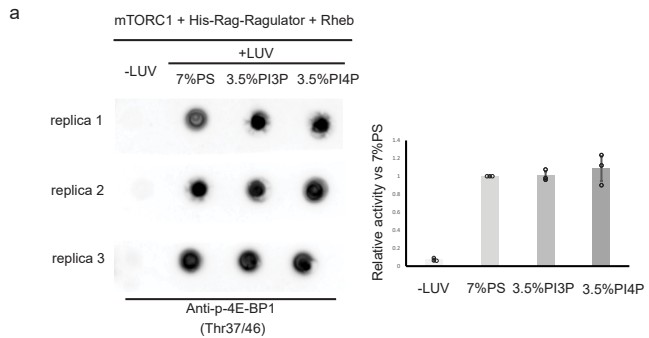

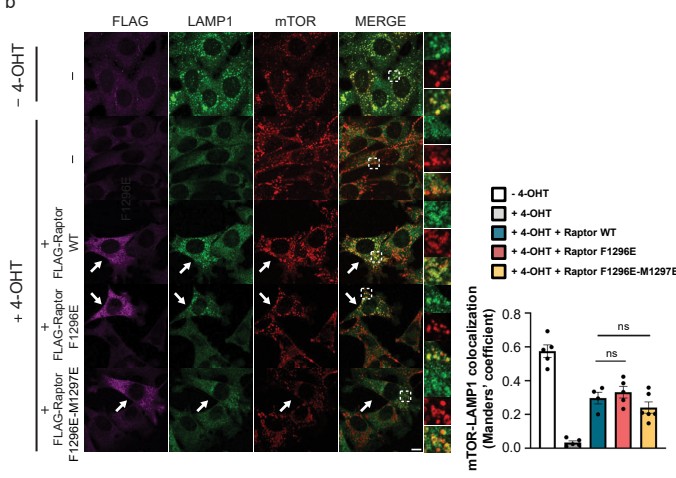

**Extended Data Fig. 6 | Effects of PI3P and PI4P on mTORC1 activity and lysosomal localization of mTOR with Raptor membrane-interaction mutants. a**, In vitro kinase assay showing mTORC1 kinase activity with different composition of negatively charged lipids. The quantification is shown for 3 repeats. **b**, iRaptor KO MEFs, untreated or treated with 0.5 μM 4-hydroxy-tamoxifen (4-OHT) for 48 h, were transfected with either WT, F1296E, or F1296/M1297E Raptor. 24 h after transfection, cells were starved of amino acids for 60 min, then re-stimulated with amino acids for 30 min. The cells were analyzed by immunofluorescence and quantified for mTOR-LAMP1 colocalization using Manders' colocalization coefficient. Results are mean ± s.e.m.; n = 3 (One-way ANOVA, Dunnett's multiple comparisons test; ns: not significant). Scale bar, 10 μm.

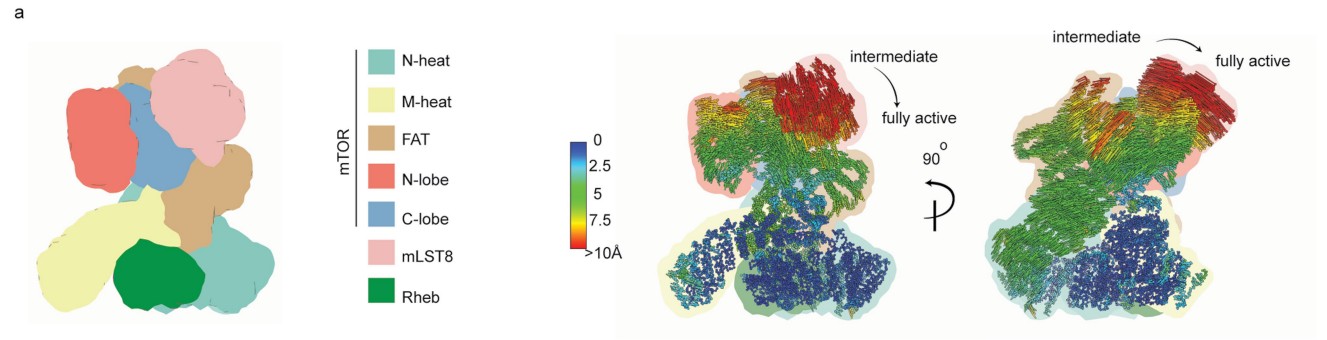

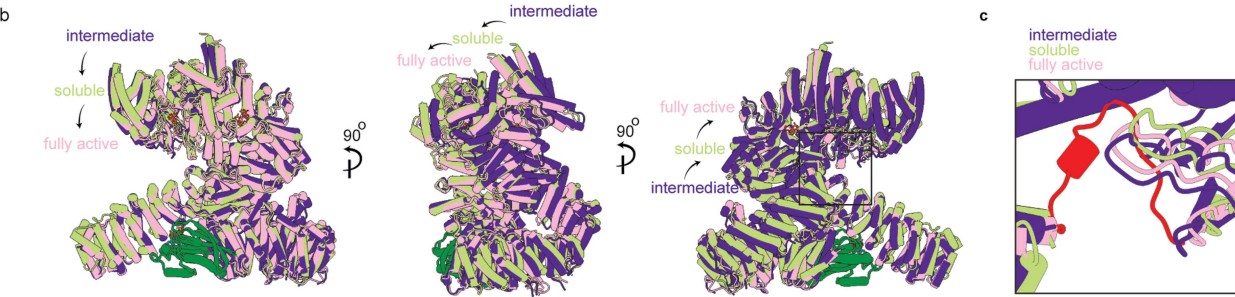

**Extended Data Fig. 7 | Structural comparison between the intermediate and fully active states of mTOR.** Structures are superimposed based on Rheb. **a**, A carton representation of the mTOR-Rheb-mLST8 subcomplex. The motion between intermediate and fully active states are indicated with lines that connect the backbone of both structures. The distance of the motion is labeled with a rainbow color. **b**, The soluble structure of mTORC1-Rheb is in between the conformational change of the intermediate and fully active states of mTORC1-Rheb on membrane. **c**, Only the fully active state contains the ordered loop in the backside of the ATP pocket. The ordered loop (904–920) is colored red.

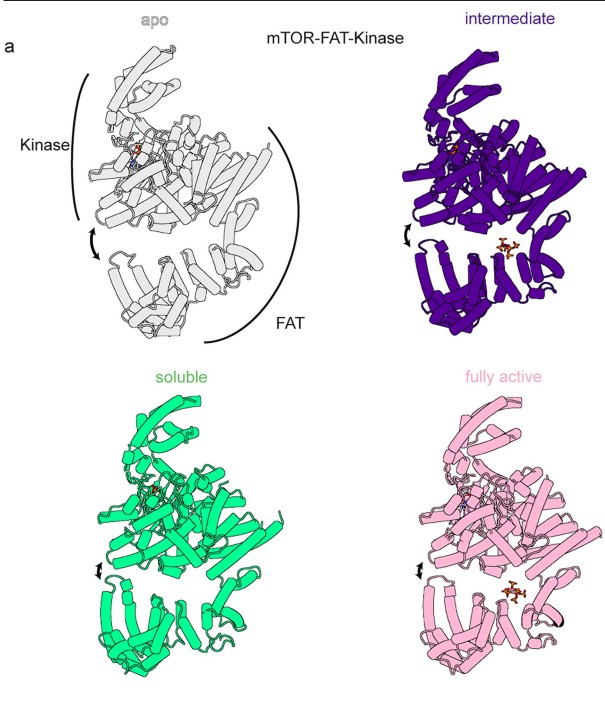

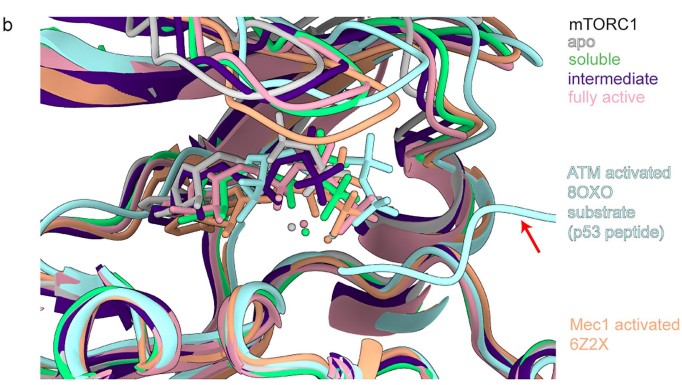

**Extended Data Fig. 8 | Comparison of the FAT-Kinase domain of mTORC1 in different states and the ATP position relative to other PIKKs. a**, The FAT (residues 1255–1453 are omitted) and kinase domains of the apo, soluble, intermediate, and fully active states of mTOR are shown side by side. The constriction between the FAT and kinase domain is indicated by double arrow curved lines. **b**, A close-up view of the ATP binding pocket of mTOR, superimposed by ATM and Mec1 based on the C-lobe (residues 2200–2400). The substrate of ATM, p53, is indicated with a red arrow.

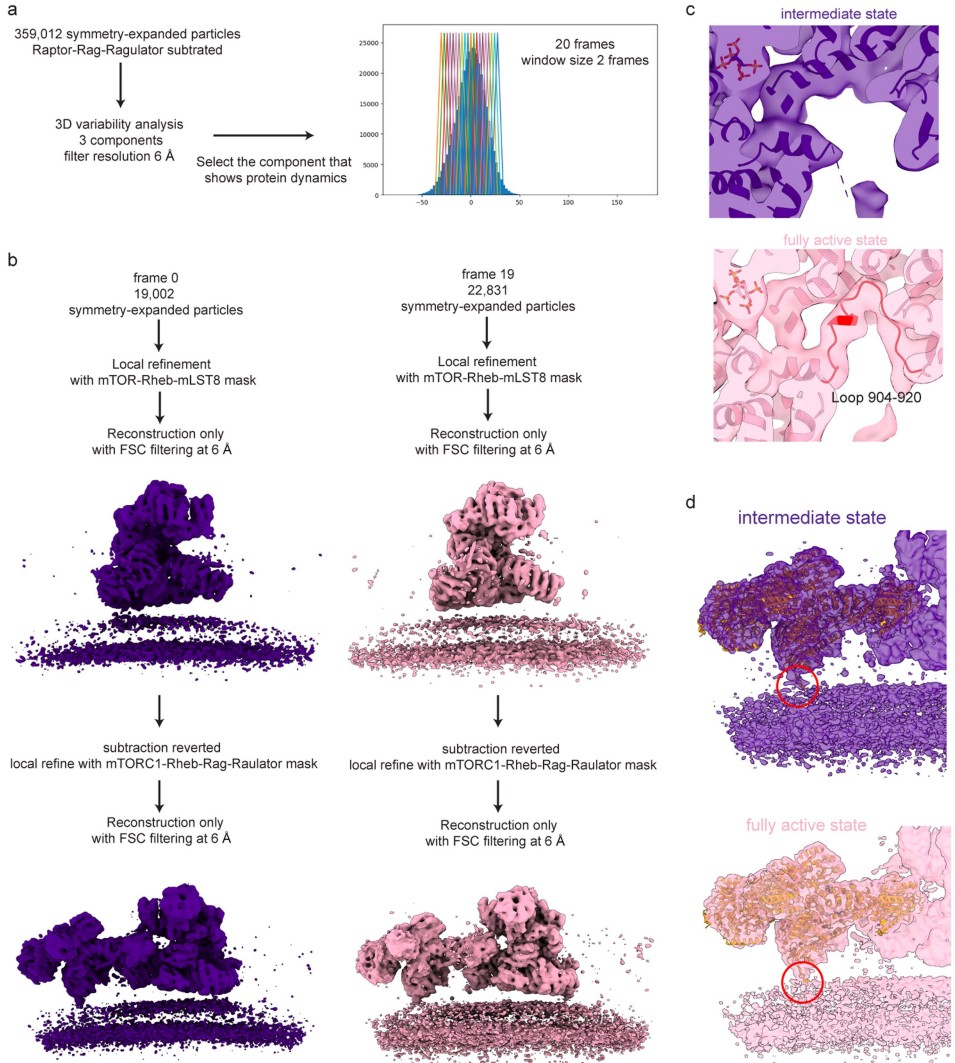

**Extended Data Fig. 9 | 3D variability analysis. a**, 3DVA processing workflow. **b**, Particles corresponding to the frames at the two ends of the continuous motion were used for local refinement. Final refinement was done with FSC filtering at 6 Å. **c**, Close-up view of the ordered loop region in the opposite of ATP binding pocket. The ordered loop is colored red. **d**, Close-up view of the intermediate and fully active states with Raptor-Rag-Ragulator subcomplex. The membrane-interacting site of Raptor is indicated by red circle.

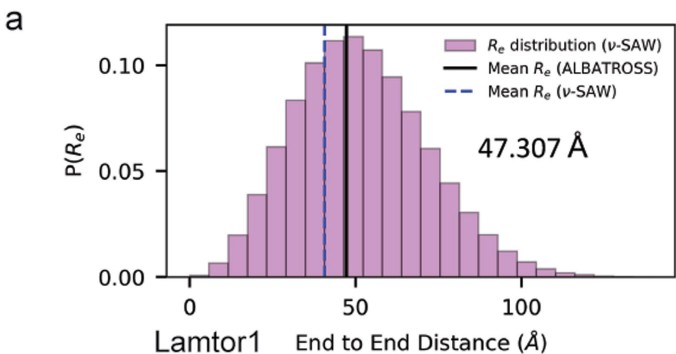

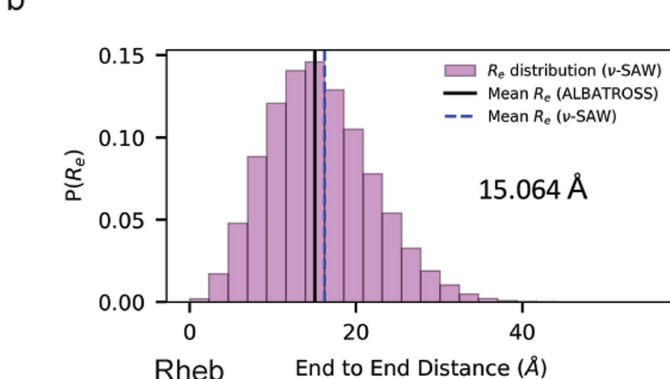

**Extended Data Fig. 10 | Predicted end-to-end distance for Lamtor1 and Rheb.** The predictions are made by the ALBATROSS-Colab (https://colab.research. google.com/github/holehouse-lab/ALBATROSS-colab/blob/main/example_ notebooks/polymer_property_predictors.ipynb). The residues from Lamtor1 (**a**, C3-A46) and Rheb (**b**, A174-S180) are used as input. The mean distance from ALBATROSS is indicated. The mean distance and distribution by $v$-SAW model are shown with dashed blue line and pink columns, respectively.

# Reporting Summary

## Statistics

For all statistical analyses, confirm that the following items are present in the figure legend, table legend, main text, or Methods section.

| n/a | Confirmed | |
|---|---|---|
| ☐ | ☒ | The exact sample size (*n*) for each experimental group/condition, given as a discrete number and unit of measurement |
| ☐ | ☒ | A statement on whether measurements were taken from distinct samples or whether the same sample was measured repeatedly |
| ☐ | ☒ | The statistical test(s) used AND whether they are one- or two-sided <br> *Only common tests should be described solely by name; describe more complex techniques in the Methods section.* |
| ☒ | ☐ | A description of all covariates tested |
| ☒ | ☐ | A description of any assumptions or corrections, such as tests of normality and adjustment for multiple comparisons |
| ☐ | ☒ | A full description of the statistical parameters including central tendency (e.g. means) or other basic estimates (e.g. regression coefficient) AND variation (e.g. standard deviation) or associated estimates of uncertainty (e.g. confidence intervals) |
| ☐ | ☒ | For null hypothesis testing, the test statistic (e.g. *F*, *t*, *r*) with confidence intervals, effect sizes, degrees of freedom and *P* value noted <br> *Give P values as exact values whenever suitable.* |
| ☒ | ☐ | For Bayesian analysis, information on the choice of priors and Markov chain Monte Carlo settings |
| ☒ | ☐ | For hierarchical and complex designs, identification of the appropriate level for tests and full reporting of outcomes |
| ☒ | ☐ | Estimates of effect sizes (e.g. Cohen's *d*, Pearson's *r*), indicating how they were calculated |

*Our web collection on statistics for biologists contains articles on many of the points above.*

## Software and code

Policy information about availability of computer code

| Data collection | SerialEM 4.2 |
|---|---|
| Data analysis | CryoSPARC v4.4, Chimera X 1.6, COOT 0.9, ISOLDE , Phenix 1.21.1, GraphPad Prism 10, and Fiji |

For manuscripts utilizing custom algorithms or software that are central to the research but not yet described in published literature, software must be made available to editors and reviewers. We strongly encourage code deposition in a community repository (e.g. GitHub). See the Nature Portfolio guidelines for submitting code & software for further information.

## Data

Policy information about availability of data

All manuscripts must include a data availability statement. This statement should provide the following information, where applicable:
- Accession codes, unique identifiers, or web links for publicly available datasets
- A description of any restrictions on data availability
- For clinical datasets or third party data, please ensure that the statement adheres to our policy

Structural coordinates were deposited in the PDB with accession codes 9ED4 (mTORC1-Rag-Ragultor-4EBP1), 9ED6 (mLST8-Rag-Ragultor), 9ED7 (fully active state of mTOR-Rheb), and 9ED8 (Intermediate state of mTOR-Rheb). The cryo-EM density maps were deposited in the Electron Microscopy Data Bank with accession numbers EMD-47932 (a composite map of mTORC1-Rag-Ragultor-4EBP1 on membrane), EMD-47933 (focused refinement of mLST8-Rag-Ragultor subcomplex), EMD-47934 (mTORC1-Rag-Ragulator-4EBP1 on membrane with two extra Rag-Ragulator), EMD-47935 (mTORC1-Rag-Ragulator-4EBP1 on membrane with one extra Rag-Ragulator), EMD-47936 (mTORC1-Rag-Ragulator-4EBP1 complex on membrane with C2 symmetry), EMD-47937 (focused refinement of the mTORC1-Rag-Ragulator-4EBP1 on membrane with mTOR-mLST8-Rheb mask), EMD-47938 (focused refinement of the mTORC1-Rag-Ragulator-4EBP1 on membrane with Raptor-Rag-Ragulator mask), EMD-47939 (fully active state of mTOR on membrane), and EMD-47940 (intermediate state of mTOR on membrane).

# Field-specific reporting

Please select the one below that is the best fit for your research. If you are not sure, read the appropriate sections before making your selection.

☒ Life sciences ☐ Behavioural & social sciences ☐ Ecological, evolutionary & environmental sciences

For a reference copy of the document with all sections, see nature.com/documents/nr-reporting-summary-flat.pdf

# Life sciences study design

All studies must disclose on these points even when the disclosure is negative.

| Sample size | a common practice of 3 replicates for in vitro biochemical assay is chosen |
|---|---|
| Data exclusions | no data were excluded from the analyses |
| Replication | all attempts at replication were successful |
| Randomization | different aliquots of protein samples from different purifications are randomly chosen for biochemical assays. |
| Blinding | Blinding is not possible because we need to know the exact components of each biochemical assay. |

# Reporting for specific materials, systems and methods

We require information from authors about some types of materials, experimental systems and methods used in many studies. Here, indicate whether each material, system or method listed is relevant to your study. If you are not sure if a list item applies to your research, read the appropriate section before selecting a response.

### Materials & experimental systems

| n/a | Involved in the study |
|---|---|
| ☐ | ☒ Antibodies |
| ☐ | ☒ Eukaryotic cell lines |
| ☒ | ☐ Palaeontology and archaeology |
| ☒ | ☐ Animals and other organisms |
| ☒ | ☐ Human research participants |
| ☒ | ☐ Clinical data |
| ☒ | ☐ Dual use research of concern |

### Methods

| n/a | Involved in the study |
|---|---|
| ☒ | ☐ ChIP-seq |
| ☒ | ☐ Flow cytometry |
| ☒ | ☐ MRI-based neuroimaging |

## Antibodies

| Antibodies used | Phospho-p70 S6 Kinase (Thr389) (1A5) (Mouse mAb, Cat# 9206 - 1:1000 WB), p70 S6 Kinase (Rabbit, Cat# 9202 - 1:1000 WB), 4E-BP1 (Rabbit, Cat# 9644 - 1:1000 WB), Phospho-4E-BP1 (Ser65) (Rabbit, Cat# 9456 - 1:1000 WB), Phospho-4E-BP1 (Thr37/46) (Rabbit, Cat # 236B4 - 1:10,000 WB) and Raptor (24C12) (Rabbit, Cat# 2280 - 1:1000 WB) were from Cell Signaling Technology; anti-GAPDH (6C5) (Rabbit, Cat# sc-32233 - 1:15000 WB) was from Santa Cruz; and FLAG M2 (Mouse, Cat# F1804 – 1:1000 WB ) was from Sigma Aldrich |
|---|---|
| Validation | Validations of antibodies used above can be found on manufacture's websites. https://www.cellsignal.com/products/primary-antibodies/phospho-p70-s6-kinase-thr389-1a5-mouse-mab/9206 https://www.cellsignal.com/products/primary-antibodies/p70-s6-kinase-antibody/9202 https://www.cellsignal.com/products/primary-antibodies/4e-bp1-53h11-rabbit-mab/9644 https://www.cellsignal.com/products/primary-antibodies/phospho-4e-bp1-ser65-174a9-rabbit-mab/9456 https://www.cellsignal.com/products/primary-antibodies/phospho-4e-bp1-thr37-46-236b4-rabbit-mab/2855 https://www.cellsignal.com/products/primary-antibodies/raptor-24c12-rabbit-mab/2280 https://www.scbt.com/p/gapdh-antibody-6c5?srsltid=AfmBOoqB9FWmJQBQ42uu9vnfp1jtE696jFx2sRdRMPoyOZugR2rYyYOe https://www.sigmaaldrich.com/US/en/product/sigma/f1804? utm_source=google&utm_medium=cpc&utm_campaign=22180208315&utm_content=179433150172&gad_source=1&gad_campai gnid=22180208315&gbraid=0AAAAAD8kLQQX0NQSM-GS9nfwDC6HlIBvo&gclid=CjwKCAjwkbzEBhAVEiwA4V- yqmKnx_hBy6rXA8v95W2p24fbsfKP7X3ZO-jiS71l8GJfV3qGITf7dBoCnGsQAvD_BwE |

# Eukaryotic cell lines

Policy information about cell lines

| | |
|---|---|
| Cell line source(s) | HEK293F GnTI is from UC Berkeley cell culture facility (https://bds.berkeley.edu/facilities/cell-culture); Inducible Raptor KO MEFs is a gift from Michael Hall, Univ. of Basel (https://link.springer.com/protocol/10.1007/978-1-61779-430-8_16) |
| Authentication | Cell lines were validated by morphological analysis |
| Mycoplasma contamination | Cell lines were routinely tested for absence of mycoplasma. |
| Commonly misidentified lines (See ICLAC register) | no commonly misidentified cell lines were used in the study |

