## [Peer Review File · Nature]

Structural basis for mTORC1 activation on the lysosomal membrane

Corresponding Author: Professor James Hurley

Version 0:

Reviewer comments:

Referee #1

(Remarks to the Author)

In this manuscript Cui et al report on the sp-cryoEM structure of the active mTORC1 complex bound to membranes via Rheb and Ragulator-Rag and in the presence of a canonical substrate. The structure of active mTORC1 bound to liposomal membranes confirms previously published (e.g. Cui et al Nature 2023; Anandapadamanaban et al Science 2019; Rogala KB et al Science 2019; Yip et al Mol Cell 2010) structural data on mTORC1 bound to Rheb and Rag-Ragulator with canonical or non-canonical substrates with some noticeable features: In the liposome-bound state Raptor undergoes a subtle rotational movement that aligns it with membrane-bound Rheb and reveals a previously unknown hydrophobic interaction site for lipids. Additional membrane interactions are seen with a basic patch on the N-HEAT domain of mTOR. Further 3D classification analysis revealed two subpopulations of membrane bound mTORC1 that are hypothesized to represent intermediate and active conformations of mTORC1. Two residues within the FAT domain of mTOR are identified that are disordered in the intermediate state but become ordered in the active state. Moreover, the active state may be further promoted by electrostatic interactions between the FAT domain and the C-lobe of the kinase core. Raptor-membrane and Rheb-mTOR interactions are preserved in both the intermediate and the active states, suggesting that the Raptor and mTOR membrane interactions described in this study may not be required for lysosomal targeting of mTORC1 per se, but rather promote its activation. Based on these results the authors propose a 3-step model for mTORC1 recruitment and activation at membranes.

The main novelty of the paper lies with the observation that membrane-bound mTORC1 undergoes a small rotational movement that aligns alleged lipid binding sites on Raptor and mTOR on a planar membrane surface, thereby, promoting phosphorylation of a canonical substrate. As it stands, I do not regard the manuscript as a strong candidate for publication in Nature. The progress reported here - while being of interest to biochemists studying mTORC1 signaling - in my view does not provide the conceptual advance expected from a Nature paper. Moreover, the functional-mechanistic analysis of the identified membrane binding sites on mTORC1 remains preliminary and somewhat superficial. Finally, it is unclear whether and how direct lipid binding to mTORC1, and by extension mTORC2, which harbors at least the basic patch on the N-HEAT domain of mTOR, affects the efficacy of phosphorylation of canonical and non-canonical substrates. The authors seem to favor the idea that lipid binding specifically affects canonical substrate phosphorylation via the Rheb axis, but no experimental evidence is provided for this assumption.

Specific points:

1. The authors hypothesize that the newly identified membrane binding sites on Raptor and mTOR-N-HEAT contribute to mTORC1 activation but no data regarding their relevance for mTORC1 localization are provided. What is the effect of mutational inactivation of the lipid binding sites on Raptor and the N-HEAT domain of mTOR on the localization of the complex in cells lacking or depleted of the endogenous proteins under different physiological conditions, e.g. steady-state or refed with amino acids \pm serum?
2. The authors have used LUVs with a comparably simple lipid composition comprising PC, PS and cholesterol but no other more specific lipids characteristic of the lysosomal limiting membrane. Do Raptor and/ or mTOR associate with specific membrane lipids and how does lipid composition impact mTORC1 activity in vitro?

3. Throughout the study the authors assess mTORC1 activity essentially using 4E-BP1 as a single canonical substrate. It seems mandatory to assess the effect of the identified lipid-binding sites on Raptor and mTOR but also the alleged FAT/C-lobe interaction at the functional level in vitro and in living cells by KO/rescue experiments.

4. A possibly significant observation of this work is the assignment of so-called intermediate and active states of membrane-associated mTORC1. The latter appears to involve interactions between the FAT and C-lobe on the kinase core via electrostatics. Surprisingly, no effort is made to functionally interrogate the physiological relevance of these interactions for mTORC1 activity with respect to canonical vs non-canonical substrates in vitro and in vivo.

5. What is the role of the novel Rag-Ragulator binding site on mLST8 with respect to mTORC1 nanoscale localization and mTOR kinase activity towards canonical and non-canonical substrates?

6. Does mutant Rheb Y35N bypass the requirement for lipid binding to mTORC1 as one might expect?

7. Ext Data Fig 9e: It is difficult to judge from the low-resolution slab representation whether or not the membrane interaction of Raptor is truly conserved in the intermediate state as no distances or specific interactions between the proposed FM finger and lipids are revealed.

Minor:

8. The manuscript needs to be carefully edited to ascertain that all refs to figures are accurate.

Referee #2

(Remarks to the Author)

The paper investigates the mechanistic target of rapamycin complex 1 (mTORC1) and how it integrates signals from growth factors and nutrients through Rheb and Rag small GTPases. Based on the hypothesis that the membrane itself has a key role, the authors have reconstituted mTORC1 activation in vitro on a LUV membrane environment. They report a cryo-electron microscopy (cryo-EM) structure that reveals direct interactions between Raptor, mTOR, and the membrane, suggesting steps that are essential for the full activation of mTORC1. The study enlightens the process of mTORC1 activation involving sequential recruitment and membrane engagement, offering insights into how growth factor and nutrient signals could be integrated at the lysosome. While previous research has established the roles of Rheb and Rag GTPases in mTORC1 activation, this paper advances our understanding by detailing the precise sequence and structural basis of their concerted function.

The investigation is well conducted and in my opinion it inherits publication in a high impact journal.

I suggest some minor corrections:

The discussion could benefit for some re-writing. The authors summarize well the findings there a but as a reader I feel like I am missing something – what are the implications of the finding to the field? For instance, could mtor within mTORC2 to interact to the plasma membrane in a similar way? Could they speculate how to probe the significance of the new Rag-Ragulator mLST8 site? I would also have expected to read some caveats of the study here – since the geometric precision is highlighted in the text, is there a risk the reconstituted system does not reflect completely the physiology? E.g could the tag recruitment, tethering etc have an effect here?

I would suggest to change “more “and “less active” conformations to something more defined. Maybe partially active and fully active?

There is Typo on Fig 5 caption -

The insect shows the ATP binding site of mTORC1 in the apo and active states replac to insect = inset

Referee #3

(Remarks to the Author)

The manuscript of Ciu et al. describes the structure of a Rag/Ragulator/RHEB/mTORC1 complex on lipid vesicles. Overall, the manuscript is an excellent contribution that would be highly appropriate to Nature readership. mTORC1 integrates growth factor signalling with amino-acid availability, so that the complex is maximally activated only in the presence of regulators associated with both signals. The RagA/RagC/Ragulator complex acquires its active form in response to amino acid availability and recruits mTORC1 to lysosomal membranes. The small G-protein RHEB is stabilized in its GTP-bound form in response to growth factor signalling and acts as an allosteric activator of mTORC1. However, mTORC1 functions in cells as an AND gate and only becomes fully active when associated with both active RHEB and active RAGs on membranes.

In vitro studies have previously shown that the effective concentration of soluble RHEB-GTP necessary for half-maximum activation of mTORC1 is around 100 micromolar. Because the concentration of RHEB in the cytosol has been estimated at

less than 1 micromolar, it has previously been suggested that RHEB might only activate mTORC1 in cells if its effective concentration is increased by co-localisation of mTORC1 and RHEB on the same membrane. Ciu et al carried out a cell-free reconstitution at a concentration of RHEB less than the cellular concentration and showed that there was significant activation of mTORC1 only in the presence of vesicles that had both membrane-coupled active Rag/Ragulator and membrane-coupled active RHEB. This reconstitution resulted in about 30-fold mTORC1 stimulation. This alone is an excellent contribution to the field. This could be further strengthened by comparing the k_{cat} of mTORC1 fully occupied with soluble RHEB-GTP (i.e., in the presence of soluble RHEB at a concentration greater than its EC_{50}) with the k_{cat} measured for the LUV/Rag-Rag/Ragulator/RHEB reconstitution. This would help address the mechanism of activation on membranes, i.e., it would clarify whether there is a unique allosteric component of mTORC1 activation conferred by the presence of membranes that is not conferred by saturating concentrations of soluble RHEB (greater than the EC_{50} of 100 micromolar) in the absence of membranes.

The bulk of the manuscript is devoted to describing the detailed anatomy of the mTORC1 complex on membranes, and the authors compare the conformation that they observe with the previously reported structure of mTORC1 bound to RHEB-GTP in the absence of membranes. This is an important component of the manuscript, but in its current form, the manuscript has not established whether these membrane-associated conformational changes have any consequence on the enzyme activity. The k_{cat} measurements suggested above might help to strengthen the link between the observed conformational changes and enzyme activation. Even if the authors feel that the k_{cat} measurements are beyond the scope of the current work, they should comment on this mechanistic issue in their discussion.

The authors show the regions of mTORC1 that associate with membranes include basic residues in N-HEAT and residues in the RAPTOR subunit. The authors made mutations of the membrane interacting residues and showed that both types of mutations affect activity. This is about all that they can do to validate the structural observations, but it would be helpful to know the effect of these mutations on the activity of mTORC1 in the absence of membranes. Ideally, they should evaluate specific enzyme activities either in the absence of RHEB or in the presence of RHEB at a concentration in excess of its EC_{50} .

Minor points

1. In the abstract, the authors state that their work establishes “a three-step process, consisting of (1) Rag-Ragulator-driven recruitment to within ~100 Angstroms of the lysosomal membrane, (2) Rheb-driven recruitment to within ~40 Angstroms, and finally (3) direct engagement of mTOR and Raptor with the membrane.” There are three things that can bind to mTORC1 in their reconstruction: Rag/Ragulator, RHEB and membranes, but if there is an order to these binding events, it is most likely due to the relative affinities. The RAPTOR/Rag association is tightest, while RHEB and direct membrane interaction are weaker. The process described by the authors seems to focus on the distance of the binders from the membrane, which is probably less important. Some discussion of this might be helpful.

2. The authors made vesicles using a range of filters. It is a common observation that the vesicle sizes do not actually agree very well with the pore size of the filters. Have the authors tried to measure the average sizes using light scattering or cryo-EM? The results that they present clearly show an effect of the vesicle size on the activity, but one of the strong points of the paper is that the authors are attempting to go beyond qualitative descriptions.

3. Extended Fig. 8b – the Apo mTORC1 model seems to be missing in this panel.

4. In the legend for Fig 1b, it would be good to say that the fold of activation is relative to the lane 1 containing only mTORC1.

Version 1:

Reviewer comments:

Referee #1

(Remarks to the Author)

In their revised manuscript the authors have responded to most of my queries satisfactorily. I also concur with the authors that their study partly answers the question which role the membrane plays in mTORC1 activation, although I am not certain that the role of specific membrane lipids in this process can be ruled out as the authors' response implies. Previous work had in particular suggested a role for PI3,5P2 in membrane recruitment and activation of mTORC1, e.g. Hasegawa et al (2022) PMID: 35020443, which I was referring to in my initial report. That said, I am OK with leaving this point for future studies.

Several points in my view still remain to be addressed before publication of this study:

- I remain of the opinion that the authors should address how mutational inactivation of the lipid binding sites on Raptor and mTOR affect canonical vs non-canonical substrate phosphorylation and see no reason why this would go beyond the scope of the study as the key proposal by the same authors has been that mTORC1 needs to be bound to the lysosomal surface via RagC to phosphorylate TFEB. Hence, assessing TFEB phosphorylation in cells expressing mutant versions of Raptor or mTOR would be an important piece of information.

- I do not agree that mutations in mTOR cannot be analyzed due to lethality of KO lines. There are numerous studies, in which mutant versions of mTOR have been expressed in WT cells or cells depleted of the endogenous mTOR protein and rescued by an si/shRNA resistant copy. The following two questions hence should be addressed:

- What is the effect of mutational inactivation of the lipid binding sites on the N-HEAT domain of mTOR on the localization of the complex in cells lacking or depleted of the endogenous proteins under different physiological conditions, e.g. steady-state or refed with amino acids \pm serum with respect to both canonical and non-canonical substrates?

- How do interactions between the FAT and C-lobe on the kinase core of mTOR affect canonical and non-canonical substrate phosphorylation?

Referee #2

(Remarks to the Author)

Cui and co-authors did a good job addressing the reviewers questions and explaining why some of the points are out of scope for this particular author. I think the changes to the manuscript make it clearer and I reiterate this study is of interest to the field and more generally enriches our understanding of the often under appreciated role of membrane context on function. I recommend this paper for publication.

Referee #3

(Remarks to the Author)

The authors have addressed all of my concerns. I also believe that the authors have addressed concerns of the other reviewers where they are within the reasonable scope of the manuscript.

Referee #1 (Remarks to the Author):

In this manuscript Cui et al report on the sp-cryoEM structure of the active mTORC1 complex bound to membranes via Rheb and Ragulator-Rag and in the presence of a canonical substrate. The structure of active mTORC1 bound to liposomal membranes confirms previously published (e.g. Cui et al Nature 2023; Anandapadamanaban et al Science 2019; Rogala KB et al Science 2019; Yip et al Mol Cell 2010) structural data on mTORC1 bound to Rheb and Rag-Ragulator with canonical or non-canonical substrates with some noticeable features: In the liposome-bound state Raptor undergoes a subtle rotational movement that aligns it with membrane-bound Rheb and reveals a previously unknown hydrophobic interaction site for lipids. Additional membrane interactions are seen with a basic patch on the N-HEAT domain of mTOR. Further 3D classification analysis revealed two subpopulations of membrane bound mTORC1 that are hypothesized to represent intermediate and active conformations of mTORC1. Two residues within the FAT domain of mTOR are identified that are disordered in the intermediate state but become ordered in the active state. Moreover, the active state may be further promoted by electrostatic interactions between the FAT domain and the C-lobe of the kinase core. Raptor-membrane and Rheb-mTOR interactions are preserved in both the intermediate and the active states, suggesting that the Raptor and mTOR membrane interactions described in this study may not be required for lysosomal targeting of mTORC1 per se, but rather promote its activation. Based on these results the authors propose a 3-step model for mTORC1 recruitment and activation at membranes.

The main novelty of the paper lies with the observation that membrane-bound mTORC1 undergoes a small rotational movement that aligns alleged lipid binding sites on Raptor and mTOR on a planar membrane surface, thereby, promoting phosphorylation of a canonical substrate.

The main novelty is the structural and biochemical elucidation of the role of the membrane in mTORC1 activation and the mechanism of signal integration between the Rag and Rheb pathways. These are fundamental and heretofore structurally unexplained aspects of mTORC1 biology. We are sorry that this message did not come through to the reviewer. In the hopes of making the message clearer, we have overhauled Fig. 5 to make the insight even more explicit.

As it stands, I do not regard the manuscript as a strong candidate for publication in Nature. The progress reported here - while being of interest to biochemists studying mTORC1 signaling - in my view does not provide the conceptual advance expected from a Nature paper. Moreover, the functional-mechanistic analysis of the identified membrane binding sites on mTORC1 remains preliminary and somewhat superficial. Finally, it is unclear whether and how direct lipid binding to mTORC1, and by extension mTORC2, which harbors at least the basic patch on the N-HEAT domain of mTOR, affects the efficacy of phosphorylation of canonical and non-canonical substrates. The authors seem to favor the idea that lipid binding specifically affects canonical substrate phosphorylation via the Rheb axis, but no experimental evidence is provided for this assumption.

Specific points:

1. The authors hypothesize that the newly identified membrane binding sites on Raptor and mTOR-N-HEAT contribute to mTORC1 activation but no data regarding their relevance for mTORC1 localization are provided. What is the effect of mutational inactivation of the lipid binding sites on Raptor and the N-HEAT domain of mTOR on the localization of the complex in cells lacking or depleted of the endogenous proteins under different physiological conditions, e.g. steady-state or refed with amino acids \pm serum?

We have now examined the localization of mTOR with the Raptor mutants using the iRaptor cell line. We did not observe differences in mTOR co-localization with lysosome between wild type Raptor and Raptor mutants. The result is as expected because the localization of mTORC1 is driven by the

interactions between Raptor and Rags. We have included these new results in a new Extended Data Figure 6, and added the following sentence in the Results section: "As expected given that these Raptor mutants are competent to bind to RagA, the localization of mTOR to lysosome was not affected by the membrane-anchoring mutations (Extended Data Fig. 6b)." It is not feasible to test the mTOR mutants in cells since mTOR is essential for cell viability, and in any event the same negative outcome would be expected for this experiment as seen for the Raptor mutants.

2. The authors have used LUVs with a comparably simple lipid composition comprising PC, PS and cholesterol but no other more specific lipids characteristic of the lysosomal limiting membrane. Do Raptor and/ or mTOR associate with specific membrane lipids and how does lipid composition impact mTORC1 activity in vitro?

In light of Ebner et al. (2023), ref. 26, which like this reviewer, proposed that specific lysosomal lipid changes modulate mTORC1 activity, we tested the effect of PI3P and PI4P toward mTORC1 activity in vitro and found that there is no difference among PS, PI3P and PI4P lipids. We have included the results in the new Extended Data Fig. 6 and add the following description in the results on pg 5-6.

" Following a report that phosphatidylinositol 3-phosphate and phosphatidylinositol 4-phosphate have opposing effects on mTORC1 activity in cells ²⁶, we tested these lipids in the *in vitro* kinase assay (Extended Data Fig. 6a) and found no difference in activity between these two phosphoinositides and another anionic lipid, phosphatidylserine. These data show that membrane curvature, but not phosphoinositide identity, directly modulates mTORC1 activity on the membrane surface."

3. Throughout the study the authors assess mTORC1 activity essentially using 4E-BP1 as a single canonical substrate.

S6K phosphorylation was also assessed in cells, see Fig. 3h, which should resolve the concern that the findings might only be relevant only for one substrate.

It seems mandatory to assess the effect of the identified lipid-binding sites on Raptor and mTOR but also the alleged FAT/ C-lobe interaction at the functional level in vitro and in living cells by KO/ rescue experiments.

This thought continues into point 4, so please see our response below.

4. A possibly significant observation of this work is the assignment of so-called intermediate and active states of membrane-associated mTORC1. The latter appears to involve interactions between the FAT and C-lobe on the kinase core via electrostatics. Surprisingly, no effort is made to functionally interrogate the physiological relevance of these interactions for mTORC1 activity with respect to canonical vs non-canonical substrates in vitro and in vivo.

As a fundamental aspect of mTOR architecture, this is a general and long-standing question in the mTOR structural biology field. While interesting, we feel it is not a new question that is raised by our study and it is not directly pertinent to the membrane context. Therefore, we view this matter as rather far removed from the scope of the study. Further, mTOR KO cells are not viable, which presents a technical obstacle to the proposed in cell experiments.

5. What is the role of the novel Rag-Ragulator binding site on mLST8 with respect to mTORC1 nanoscale localization and mTOR kinase activity towards canonical and non-canonical substrates?

Our view is the RagA-mLST8 interaction is "coming along for the ride", perhaps supplementing the main recruitment interaction in a small but phenotypically indiscernible, or perhaps having a secondary effect by modulating PRAS40 regulation. The Sabatini lab showed in 2006 (PMID 17141160) that mLST8 is required only for mTORC2 activity and is dispensable for mTORC1. This was corroborated by others (PMID 31085701). While there might be small effects on activity or PRAS40 regulation found by expressing versions of mTORC1 lacking mLST8, given the difficulty in demonstrating biological importance, these details are not germane to the main advance and we did not think it made sense to invest the substantial effort involved in recruiting mLST8-deficient mTORC1. We are open to moving the description of the extra RagA interaction to the extended data if there is a need to save space.

6. Does mutant Rheb Y35N bypass the requirement for lipid binding to mTORC1 as one might expect?

We conducted the in vitro kinase assay using Rheb Y35N mutant. However, we did not observe the increased mTORC1 activity between Rheb Y35N compared to the wild type at various concentrations.

Therefore, we have condensed the discussion regarding the Rheb Y35N mutation to one sentence on pg. 8, " In this light, it is interesting that the mutation *RHEB* Y35N is linked to cancer³¹ and expression of Rheb Y35N hyperactivates mTORC1^{32,33}". The relationship between the structural shift in Tyr35 and the

cancer phenotype is intriguing but will need future work to clarify, which is beyond the scope of this paper.

7. Ext Data Fig 9e: It is difficult to judge from the low-resolution slab representation whether or not the membrane interaction of Raptor is truly conserved in the intermediate state as no distances or specific interactions between the proposed FM finger and lipids are revealed.

In the revised ED Fig. 9e, note the region indicated by the circle. Although the finger density of the intermediate state is weaker than that of the active state, both densities are visible and can be seen to directly contact the membrane.

Minor:

8. The manuscript needs to be carefully edited to ascertain that all refs to figures are accurate.

Done

Referee #2 (Remarks to the Author):

The paper investigates the mechanistic target of rapamycin complex 1 (mTORC1) and how it integrates signals from growth factors and nutrients through Rheb and Rag small GTPases. Based on the hypothesis that the membrane itself has a key role, the authors have reconstituted mTORC1 activation in vitro on a LUV membrane environment. They report a cryo-electron microscopy (cryo-EM) structure that reveals direct interactions between Raptor, mTOR, and the membrane, suggesting steps that are essential for the full activation of mTORC1. The study enlightens the process of mTORC1 activation involving sequential recruitment and membrane engagement, offering insights into how growth factor and nutrient signals could be integrated at the lysosome. While previous research has established the roles of Rheb and Rag GTPases in mTORC1 activation, this paper advances our understanding by detailing the precise sequence and structural basis of their concerted function.

The investigation is well conducted and in my opinion it inherits publication in a high impact journal.

We are very grateful for the reviewer's enthusiasm and appreciation of the advance.

I suggest some minor corrections:

The discussion could benefit for some re-writing. The authors summarize well the findings there a but as a reader I feel like I am missing something – what are the implications of the finding to the field?

Fig. 5 and the associated discussion has been redone in order to better tie the various observations in this manuscript and previous literature into an integrated model.

For instance, could mtor within mTORC2 to interact to the plasma membrane in a similar way?

We superimposed mTORC2 (cyan) structure on our membrane-bound mTORC1 structure by overlaying one mTOR subunit and found that only one mTOR subunit of mTORC2 complex could potentially interact with membrane. Without some large conformational change, it seems that mTOR subunits in mTORC2 wouldn't be able to interact with membrane in a similar way as in mTORC1. An experimental structure determination of activated mTORC2 on a membrane would likely be needed to obtain further insight, which is clearly beyond the scope of this study. In the absence of data, it seems better not to add speculation on the topic to the discussion.

Could they speculate how to probe the significance of the new Rag-Ragulator mLST8 site?

As noted also in response to Rev. 1, our view is the RagA-mLST8 interaction is "coming along for the ride" or perhaps supplementing the main recruitment interaction in a small but phenotypically indiscernible. While there might be small effects on activity found by expressing versions of mTORC1 lacking mLST8, given the difficulty in demonstrating biological importance, we did not think it made sense to invest the substantial effort involved in recruiting mLST8-deficient mTORC1. The Sabatini lab showed in 2006 (PMID 17141160) that mLST8 is required only for mTORC2 activity (which would of course not involve RagA) and is dispensable for mTORC1. We have added this citation.

I would also have expected to read some caveats of the study here – since the geometric precision is highlighted in the text, is there a risk the reconstituted system does not reflect completely the physiology? E.g could the tag recruitment, tethering etc have an effect here?

We added this sentence to the discussion: " While the addition of the membrane context greatly increases the realism of the system as compared to past membrane-free structural analyses, it is of course possible that yet further changes will be seen once it becomes feasible to reach atomistic resolution in situ and determine the structure as found on lysosomes within cells."

I would suggest to change "more "and "less active" conformations to something more defined. Maybe partially active and fully active?

We have adopted the term "fully active" to more clearly distinguish the more active state from the "intermediate" state.

There is Typo on Fig 5 caption -

The insect shows the ATP binding site of mTORC1 in the apo and active states ◇ replac to insect = inset

Fixed

Referee #3 (Remarks to the Author):

The manuscript of Ciu et al. describes the structure of a Rag/Ragulator/RHEB/mTORC1 complex on

lipid vesicles. Overall, the manuscript is an excellent contribution that would be highly appropriate to Nature readership.

We are very grateful for the reviewer's enthusiasm.

mTORC1 integrates growth factor signalling with amino-acid availability, so that the complex is maximally activated only in the presence of regulators associated with both signals. The RagA/RagC/Ragulator complex acquires its active form in response to amino acid availability and recruits mTORC1 to lysosomal membranes. The small G-protein RHEB is stabilized in its GTP-bound form in response to growth factor signalling and acts as an allosteric activator of mTORC1. However, mTORC1 functions in cells as an AND gate and only becomes fully active when associated with both active RHEB and active RAGs on membranes.

In vitro studies have previously shown that the effective concentration of soluble RHEB-GTP necessary for half-maximum activation of mTORC1 is around 100 micromolar. Because the concentration of RHEB in the cytosol has been estimated at less than 1 micromolar, it has previously been suggested that RHEB might only activate mTORC1 in cells if its effective concentration is increased by co-localisation of mTORC1 and RHEB on the same membrane. Ciu et al carried out a cell-free reconstitution at a concentration of RHEB less than the cellular concentration and showed that there was significant activation of mTORC1 only in the presence of vesicles that had both membrane-coupled active Rag/Ragulator and membrane-coupled active RHEB. This reconstitution resulted in about 30-fold mTORC1 stimulation. This alone is an excellent contribution to the field.

We thank the reviewer for their appreciation of the biochemical aspect of the story.

This could be further strengthened by comparing the k_{cat} of mTORC1 fully occupied with soluble RHEB-GTP (i.e., in the presence of soluble RHEB at a concentration greater than its EC_{50}) with the k_{cat} measured for the LUV/Rag-Rag/Ragulator/RHEB reconstitution. This would help address the mechanism of activation on membranes, i.e., it would clarify whether there is a unique allosteric component of mTORC1 activation conferred by the presence of membranes that is not conferred by saturating concentrations of soluble RHEB (greater than the EC_{50} of 100 micromolar) in the absence of membranes. The bulk of the manuscript is devoted to describing the detailed anatomy of the mTORC1 complex on membranes, and the authors compare the conformation that they observe with the previously reported structure of mTORC1 bound to RHEB-GTP in the absence of membranes. This is an important component of the manuscript, but in its current form, the manuscript has not established whether these membrane-associated conformational changes have any consequence on the enzyme activity. The k_{cat} measurements suggested above might help to strengthen the link between the observed conformational changes and enzyme activation. Even if the authors feel that the k_{cat} measurements are beyond the scope of the current work, they should comment on this mechanistic issue in their discussion.

We appreciate that the referee sees the enzyme kinetic analysis as potentially beyond the scope of the study. We would agree. Moreover, we are skeptical that bulk k_{cat} measurements alone would add enough to the picture to warrant the effort. Making an incisive linkage between the complex sequence of structural changes as depicted in the model in Fig. 5 vis-à-vis kinetics would in our view require three-color single molecule FRET analysis. This will be specialized and very technically demanding, best executed as a long-term project by an expert single molecule lab. Given the short range of some of the key motions, we are not sure that even smFRET will be able to corroborate the linkages. We added a penultimate sentence in the discussion that said " Further kinetic investigation of this model by methods such as single molecule FRET probes of might further advance understanding of the relationship between enzyme kinetics on the one hand and the movements at various size scales on the other."

The authors show the regions of mTORC1 that associate with membranes include basic residues in N-HEAT and residues in the RAPTOR subunit. The authors made mutations of the membrane interacting residues and showed that both types of mutations affect activity. This is about all that they can do to validate the structural observations, but it would be helpful to know the effect of these mutations on the activity of mTORC1 in the absence of membranes. Ideally, they should evaluate specific enzyme activities either in the absence of RHEB or in the presence of RHEB at a concentration in excess of its EC50.

We have evaluated the kinase activity of Raptor and mTOR membrane-binding mutants in the absence of membrane with or without Rheb. The enzyme activities of mTORC1 mutants are similar to the wild type with Rheb at a concentration of 116 μ M, as expected since the mutated residues have no functional role in the absence of membranes.

Minor points

1. In the abstract, the authors state that their work establishes “a three-step process, consisting of (1) Rag-Ragulator-driven recruitment to within \sim 100 Angstroms of the lysosomal membrane, (2) Rheb-driven recruitment to within \sim 40 Angstroms, and finally (3) direct engagement of mTOR and Raptor with the membrane.” There are three things that can bind to mTORC1 in their reconstruction: Rag/Ragulator, RHEB and membranes, but if there is an order to these binding events, it is most likely due to the relative affinities. The RAPTOR/Rag association is tightest, while RHEB and direct membrane interaction are weaker. The process described by the authors seems to focus on the distance of the binders from the membrane, which is probably less important. Some discussion of this might be helpful.

This paper is focused on structural analysis, and distances are inherent to the thinking, so we would argue that the spatially-focused discussion is appropriate.

2. The authors made vesicles using a range of filters. It is a common observation that the vesicle sizes do not actually agree very well with the pore size of the filters. Have the authors tried to measure the average sizes using light scattering or cryo-EM? The results that they present clearly show an effect of the vesicle size on the activity, but one of the strong points of the paper is that the authors are attempting to go beyond qualitative descriptions.

We performed the measurement of the diameter of liposomes after extrusion using Dynamic Light Scattering. We observed a clear trend of smaller vesicle sizes when using smaller pore size filters. The average size of liposomes for filters of 400 nm, 200 nm, 100 nm, 50 nm, and 30 nm are about

355 nm, 148 nm, 110 nm, 81 nm, and 67 nm respectively. We have modified the main figure to reflect the actual sizes of the liposomes and added the description in the method section.

3. Extended Fig. 8b – the Apo mTORC1 model seems to be missing in this panel.

We have added the Apo mTORC1 model to the panel.

4. In the legend for Fig 1b, it would be good to say that the fold of activation is relative to the lane 1 containing only mTORC1.

We have added the description in the figure legend.